# Whole-genome analysis of Nigerian patients with breast cancer reveals ethnic-driven somatic evolution and distinct genomic subtypes

Naser Ansari-Pour [1,2,24], Yonglan Zheng [3,24], Toshio F. Yoshimatsu [3], Ayodele Sanni[4], Mustapha Ajani [5], Jean-Baptiste Reynier[3], Avraam Tapinos[6], Jason J. Pitt[7], Stefan Dentro[8,9], Anna Woodard[3,10], Padma Sheila Rajagopal[3], Dominic Fitzgerald[11], Andreas J. Gruber[1,6], Abayomi Odetunde[12], Abiodun Popoola[13], Adeyinka G. Falusi[12], Chinedum Peace Babalola [14], Temidayo Ogundiran[15], Nasiru Ibrahim[16], Jordi Barretina [17], Peter Van Loo [18], Mengjie Chen [19,20], Kevin P. White[21], Oladosu Ojengbede[22], John Obafunwa[4], Dezheng Huo [23], David C. Wedge [1,6✉] & Olufunmilayo I. Olopade [3✉]

Black women across the African diaspora experience more aggressive breast cancer with higher mortality rates than white women of European ancestry. Although inter-ethnic germline variation is known, differential somatic evolution has not been investigated in detail. Analysis of deep whole genomes of 97 breast cancers, with RNA-seq in a subset, from women in Nigeria in comparison with The Cancer Genome Atlas (n = 76) reveal a higher rate of genomic instability and increased intra-tumoral heterogeneity as well as a unique genomic subtype defined by early clonal *GATA3* mutations with a 10.5-year younger age at diagnosis. We also find non-coding mutations in bona fide drivers (*ZNF217* and *SYPL1*) and a previously unreported INDEL signature strongly associated with African ancestry proportion, under-scoring the need to expand inclusion of diverse populations in biomedical research. Finally, we demonstrate that characterizing tumors for homologous recombination deficiency has significant clinical relevance in stratifying patients for potentially life-saving therapies.

A full list of author affiliations appears at the end of the paper.

Black women of African ancestry worldwide face breast cancer at younger ages, experience more clinically aggressive disease, present with more advanced disease at diagnosis and suffer higher mortality relative to women of other ancestries[1,2]. While socioeconomic and structural barriers explain some of this disparity, women of African ancestry also experience higher rates of estrogen receptor-negative (ER−) and progesterone receptor-negative (PR−) [hormone receptor-negative, HR−] or human epidermal growth factor receptor 2 (ERBB2)-amplified [HER2+] subtypes of breast cancer[3–5]. At least 40% of this subtype distribution is estimated to be due to heritable factors[6].

Studies of breast cancer genomes reveal population-specific differences in germline predisposition mutation frequency, somatic mutation landscapes, and mutational signatures, mirroring population differences in molecular subtypes[7–10]. Breast cancer patients of African ancestry demonstrate more TP53 alterations and fewer PIK3CA alterations[6,8], and Nigerian HR+/HER2− tumors are characterized by increased homologous recombination deficiency (HRD) signature[8]. Tumors from patients of African ancestry have also previously been shown to demonstrate increased intra-tumor heterogeneity (ITH)[11].

Whole genome sequencing (WGS) with paired germline tissue can be used to reconstruct the evolutionary "life history" of breast tumors, providing a detailed roadmap for early or late clonal and subclonal genomic events that help prioritize therapeutic targets[12–14]. To date, however, the evolutionary and clonal structure of breast cancers have only been derived using tumors predominantly of non-African ancestry. We hypothesized that studying the evolutionary trajectory of tumors from indigenous African women would provide insight into population-specific genomic features relevant to the breast cancer burden in previously understudied populations.

Here, we perform life history analysis on an indigenous Black African population, underscoring the critical paucity of genomic data previously available from breast cancer patients of African ancestry[15,16].

## Results

**High-depth WGS** was performed on 100 breast tumors (90 × depth; of which 49 had complementary RNA-seq) and paired normal tissue (30 × depth) from women with breast cancer from Nigeria as previously described[8]. Key events in the somatic evolution of these tumors were identified and compared with a similar analysis of WGS from 76 breast cancer cases from The Cancer Genome Atlas (TCGA). Three samples from Nigeria were excluded due to low purity estimates (<10%), resulting in a final set of 173 samples comprising Nigerian Black (Nigerian for short, $n = 97$), White TCGA (White for short, $n = 46$), and Black TCGA (Black for short, $n = 30$) groups (Supplementary Table 1 and Supplementary Fig. 1). Epidemiological risk factors for Nigerian cases are presented in Supplementary Table 2. The genetic ancestry information of breast cancer patients from TCGA was obtained from our previous study (Methods)[6].

**Detection of coding and non-coding bona fide drivers.** We observed a higher insertion and deletion (indel) burden in the Nigerian group compared with the White and Black groups ($P = 6.5 \times 10^{-5}$ and $P = 2 \times 10^{-4}$ respectively), which remained significant after adjusting for clinical subtype. However, the single nucleotide variant (SNV) rate did not significantly differ between races/ethnicities. Somatic coding drivers were identified with cDriver[17] (recurrence ≥ 2%, false discovery rate [FDR] < 0.01) and MutSigCV[18] (FDR < 0.05) independently. In total, thirteen driver genes were identified (Supplementary Table 3). Using the 20/20

principle[19], driver genes were classified into oncogenes (ONC) and tumor suppressor genes (TSG). GATA3 showed the strongest TSG signal and, of the five previously unreported driver genes detected; three showed a TSG signal (Supplementary Fig. 2). The cancer cell fraction (CCF) distribution of mutations showed that most drivers occurred clonally, i.e., in all tumor cells (Supplementary Fig. 3), with no significant difference in the CCF distribution of top mutated drivers (>10%) among the three groups (Supplementary Fig. 4). However, BCLAF1, a transcription regulator involved in DNA damage response[20] which also displayed a strong TSG score, was found to occur predominantly subclonal. Based on the union set of previously identified breast cancer drivers[8,21] and those identified here, 30 were identified in our samples, and 93.6% of all samples were mutated in at least one known driver (Fig. 1).

Driver enrichment analysis identified GATA3 as the only driver significantly enriched in the Nigerian group (FDR = 0.038, odds ratio [OR] = 6.3, 95% confidence interval [CI] 1.8–34.3). This enrichment remained significant after adjusting for clinical subtype (generalized linear model, $P = 0.0032$). Subtype stratification identified LAMB3 enriched in HER2+ tumors (OR = 13.7), although lacking significance following multiple testing correction (FDR = 0.15). LAMB3 occurred only in Nigerian HR−/HER2+ patients. TP53 was enriched in ER-patients (FDR = 0.0021, OR = 3.8, 95% CI 1.9–7.8). Interestingly, although GATA3 has been reported to be strongly enriched and unique to ER+ tumors[22], we did not observe such an enrichment (OR = 2.7, FDR = 0.17) in the Nigerian group, which included ten GATA3 mutants, ER− tumors (see Supplementary Figs. 5 and 6 for the distribution of GATA3 mutations). We also examined whether any of the ten pan-cancer canonical pathways[23] was enriched in any of the three groups. Mutation recurrence was first computed in each pathway for all three groups based on the associated genes (Supplementary Figs. 7–9) and then compared among the three groups. Although no pathway was significantly enriched in a particular group, we did observe a significant positive cline of mutation recurrence in the HIPPO pathway from White to Black to Nigerian groups (proportion trend test $P = 1.7 \times 10^{-5}$; Supplementary Fig. 10).

We identified hotspots for non-coding mutations by comparing the Nigerian, Black, and White groups. Two regions across the genome showed significant differences (FDR < 0.1) in mutation rates between the Nigerian and White groups (Fig. 2), both over-represented in the Nigerian group. No significant differences were identified between the Nigerian and Black groups. The strongest signal (42.3% [95% CI 32.6%-52.0%] versus 4.3% [95% CI 0%-10.2%], FDR = 0.037) was found at 20q13.2 where mutations clustered immediately upstream of ZNF217, a gene encoding a transcription factor which is a key regulator of tumorigenesis[24] and previously associated with clinical outcomes in breast cancer[25]. The second hotspot (28.9% versus 0%, FDR = 0.097) was found at 7q22.3 within and flanking SYPL1 (Synaptophysin-like 1). Although there is no evidence of its association with breast cancer, this gene has been previously associated with clinical outcomes in hepatocellular carcinoma[26] and pancreatic ductal adenocarcinoma[27]. Interestingly, we saw a significant positive cline in the prevalence of mutations in both genes from White to Black to Nigerian groups (proportion trend test; $P = 3.4 \times 10^{-6}$ for ZNF217 and $3.3 \times 10^{-4}$ for SYPL1), suggesting an association with African ancestry. Whole-transcriptome data were available in a subset ($n = 49$) of Nigerian WGS samples. We examined the effect of non-coding mutations on the expression of these two genes in the Nigerian and Black groups, where this comparison was statistically informative. We observed elevated expression in mutant tumors for both genes; however, none were found to be statistically significant

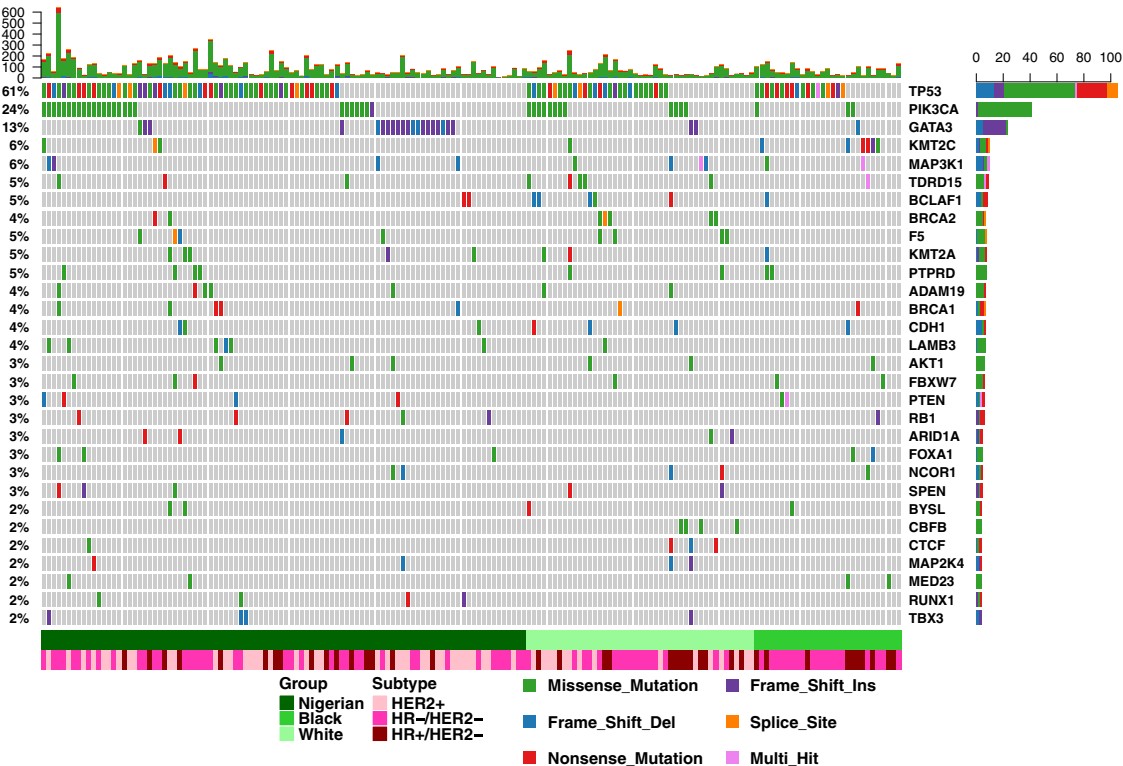

**Fig. 1 Landscape of driver genes in breast cancer across different ethnic groups.** Genes were identified using two different detection methods (cDriver and MutSigCV; n = 13). Breast cancer drivers not detected due to insufficient statistical power (n = 173 independent tumors) but frequent in this dataset (≥2%) were also added to the overall list of drivers of breast cancer (n = 30) to visualize their distribution in the Nigerian, Black, and White groups. Multi-hit: more than one non-silent variant detected in a gene in one tumor.

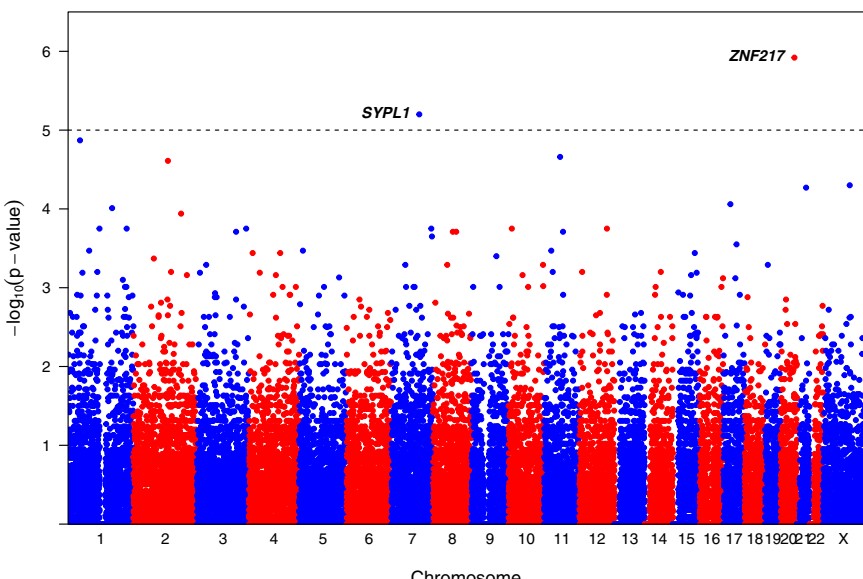

**Fig. 2 Manhattan plot for a genome-wide somatic non-coding variant enrichment analysis in the Nigerian group.** The dotted horizontal line represents the genome-wide significance threshold (pairwise Fisher's exact test two-sided P-values adjusted by Benjamini–Hochberg FDR < 0.1) with two bins showing significant enrichment of somatic non-coding variants in two previously unreported breast cancer drivers.

(Supplementary Fig. 11). The functional consequences of the identified non-coding mutations are yet to be established.

**Enrichment of a previously unreported indel mutational signature in African tumors.** Somatic mutational signatures may provide an etiological explanation for both exogenous and endogenous risk factors of breast cancer. Mutational signature analysis identified 13 single-base substitution (SBS) COSMIC signatures (Fig. 3a–b). Those observed in >5% of samples have been previously reported with a similar order of prevalence[22], with the exception of SBS39, which is a recently detected

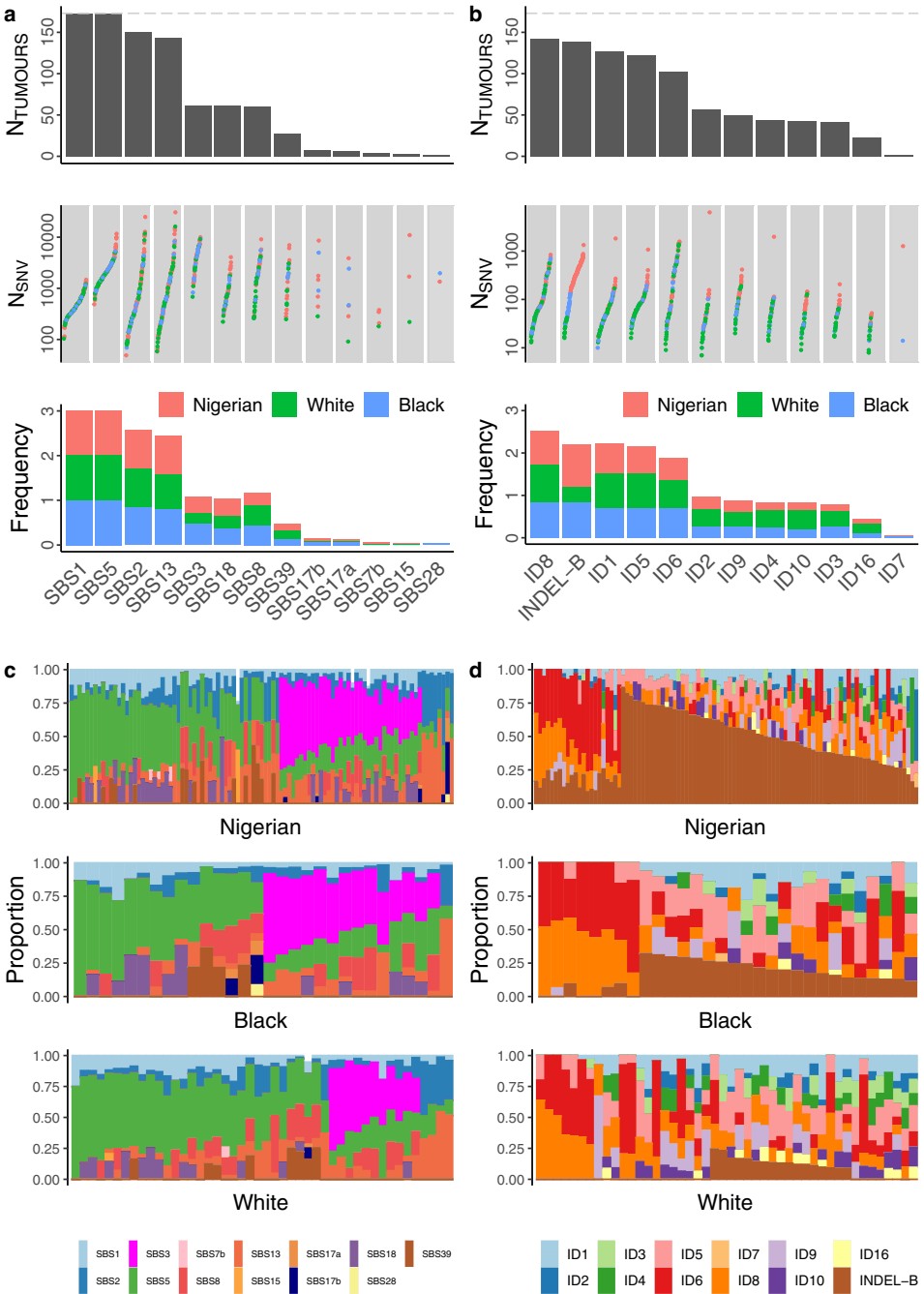

**Fig. 3 Mutational signatures in breast cancer tumors across different ethnic groups. a, b** Single-base substitution (SBS) signatures in all groups. **a** from top to bottom: number of tumors with SBS signatures across the entire dataset (the dotted line represents total sample size, $n = 173$) with signatures sorted left to right by descending frequency, number of mutations per sample (color representing groups) in respective signatures and proportion of samples carrying each signature in each group. **b** proportion of mutations assigned to each SBS signature across the three groups. **c, d** Identical plots as in **a, b** respectively for insertion/deletion (INDEL) signatures identified in the three groups. INDEL-B is a previously unreported signature characterized mainly by 5+bp insertions. Source data are provided as a Source Data file.

signature[28]. The rare SBS signatures (SBS17a/b, SBS7b, SBS15, and SBS28) were observed primarily in Nigerian and Black groups, with the mean prevalence of these signatures at 2.8 and 3.3%, respectively. The HRD signature SBS3 was observed in all groups. However, compared with the White group, the Black (OR = 2.74, $P = 0.048$) and Nigerian (OR = 1.87, $P = 0.13$) groups had slightly higher activity. Nine double-base substitution (DBS) signatures were also identified, of which five were previously unreported (Supplementary Fig. 12). Although DBS-B

was observed in similar frequencies across the groups (OR~1.3, $P > 0.58$), DBS-B showed a higher contribution in Nigerians than in the White (1.55-fold, $P = 0.0035$) and Black (1.42-fold, $P = 0.018$) groups.

Twelve INDEL signatures were detected, of which ID8 and INDEL-B were the most frequent. Of note, previously unreported signature INDEL-B was not only significantly depleted in the White group compared with the Nigerian ($P = 1.1 \times 10^{-18}$) and the Black groups ($P = 1.2 \times 10^{-4}$), but it

also showed a clear positive cline from White to Black to Nigerian groups both in prevalence (proportion trend test $P = 3.9 \times 10^{-18}$; Fig. 3c) and activity (proportion trend test $P = 2.6 \times 10^{-6}$; Fig. 3d). Moreover, unlike common INDEL signatures such as ID6, which are composed of short deletions, this signature comprises short insertions (Supplementary Fig. 13). Although the etiology of this signature remains to be elucidated, the data suggest a strong association with recent African ancestry. This association was not observed for any other INDEL signature. Notably, the indel burden in Nigerians was bimodally distributed (Supplementary Fig. 14), suggesting greater activity in a subset of patients. Assessment of the high burden samples identified ID2, ID4, and ID6 as the dominant signatures, all of which showed at least 2-fold higher mean activity than in low burden samples.

A comparison of hormone subtypes (Supplementary Fig. 15) revealed that SBS3, ID6, and ID8 were more active in the HR−/HER2− subtype while INDEL-B and DBS11 had the highest activity in the HER2+ subtype.

**Indel and structural variant (SV) signatures better classify tumors for HRD**. The HRD SBS signature (SBS3) was detected in 4/7 and 7/8 germline and somatic *BRCA*-positive tumors, respectively. While four samples with either germline or somatic *BRCA* variants lacked SBS3, all four samples exhibited high activity of INDEL signatures ID6 and ID8 (Supplementary Data 1 and Supplementary Fig. 16), both of which are associated with DSB repair by non-homologous end-joining[28]. It seems likely that, due to the high similarity of 'flat' SBS signatures (Supplementary Fig. 17), SBS3 activity may have been misassigned mainly to SBS39 during signature deconvolution, an interpretation supported by the mutual exclusivity of these two signatures (Supplementary Fig. 16).

To shed further light on the classification of tumors for HRD, we applied CHORD[29] to the entire dataset and found 34% of tumors to be HR-deficient (Fig. 4 and Supplementary Fig. 18). No significant enrichment of HRD was observed in any ethnic group or clinical subtype (Supplementary Fig. 18); however, Nigerian HR+/HER2− tumors had elevated HRD compared with Black and White groups, which is consistent with our previous finding[8]. Unlike SBS3, all *BRCA*-positive tumors were classified as HR-deficient by CHORD. Further, we observed a strong positive correlation between ID6 + ID8 signature activity and CHORD HRD score ($R = 0.93$, $P < 2.2 \times 10^{-16}$) and identified a clear cluster of tumors with high ID6 + ID8 activity (>0.5) and high CHORD score (>0.75), of which 34 (67%) were *BRCA*-negative tumors (Supplementary Fig. 19). This shows that both metrics can be used to identify HR-deficient tumors and a higher cut-off (i.e., 0.75) for CHORD score is recommended to confidently classify tumors for HRD. Next, we predicted *BRCA*-type using CHORD and examined its correlation with driver mutations. All *BRCA1*- and *BRCA2*-positive tumors were correctly classified by CHORD. Consistent with Nguyen et al.[29], HR-deficient tumors with *PALB2* were also classified as *BRCA2*-type. Of those tumors with no mutation in an HRD-associated gene but in the high-score cluster, 70% were *BRCA1*-type, which showed significant enrichment in the Nigerians compared with non-Nigerians (OR = 9.2, $P = 0.049$). In addition to CHORD, to confirm the presence of the two *BRCA*-types, SV signature analysis was undertaken, and we extracted seven SV signatures (S1–S7; Supplementary Figs. 20 and 21). Similar to Nik-Zainal et al.[22], we identified distinct *BRCA1*- and *BRCA2*-associated signatures (S3 and S2 respectively), which matched well with CHORD-based *BRCA*-type classification of tumors (94% and 84%

respectively; Fig. 4 and Supplementary Fig. 22), showing significant enrichment of S2 activity in *BRCA2*-type tumors (2.8-fold; $P = 4.5 \times 10^{-5}$) and S3 activity in *BRCA1*-type tumors (14.5-fold; $P = 1.1 \times 10^{-11}$). These results together suggest that (1) SBS3 is a poor classifier of HRD and (2) the combination of INDEL and SV signatures along with CHORD HRD score is not only superior in identifying HR-deficient tumors, but such information can also be used to infer their *BRCA*-type.

**Loss of 14q is highly enriched in HER2+ Nigerian tumors**. The CNA landscape of the Nigerian group (Fig. 5a) was very similar to that of the Black group, with all enriched CNA regions in the Nigerian group also observed in the Black group. We, therefore, compared the CNA landscape only with the White group. The key enriched CNA events unique to the Nigerian group were 5p15.33–13.3 Gain, 7p22.1–14.2 Gain, 17p13.3 Gain, and 14q LOH. We further analyzed these enriched CNAs at the clinical subtype level and found that clonal 14q LOH was highly enriched in the Nigerians in the HR−/HER2+ subtype (0.58 versus 0.07, 8.6-fold, $P = 7 \times 10^{-4}$; Fig. 5b) even though the proportion of this subtype was comparable between the two groups (43.8% and 33.3% respectively, $P = 0.358$). This enrichment was corroborated when we compared the Pan-cancer Analysis of Whole Genomes (PCAWG) HER2+ breast cancer patients of White European descent ($n = 22$; ~90% of all HER2+) with HER2+ Nigerians (3.2-fold, $P = 0.0034$).

14q LOH enrichment in the Nigerian group is an interesting observation since it is known to be associated with aggressive breast cancer progression[30,31]. Its effect may be particularly exacerbated in Nigerian patients as HR−/HER2+ has been reported to be enriched within younger Nigerian patients for reasons that are poorly understood[8]. 14q LOH has been reported in *BRCA2* mutation carriers using array-CGH[32,33]. We also observed 14q LOH in all *BRCA2* carriers; however, the presence of 14q LOH in *BRCA2*-negative tumors suggests that this CNA event is present more widely in breast cancer than previously described.

**Higher Genomic Instability (GI) in Nigerian tumors**. GI is a known hallmark of cancer which manifests as whole-genome duplication (WGD), chromosomal instability (CIN), and kataegis. We observed a 3-fold higher rate of WGD in Nigerians compared with the White group (FDR = 0.02) but no significant difference was observed between either group and the Black group. Interestingly, we observed a significant positive trend in WGD rate from White to Black to Nigerian groups (proportion trend test, $P = 0.004$).

The proportion of the genome altered adjusted with the number of CNA segments (PGAn) was calculated for all samples as a measure of CIN. We observed a 1.8-fold higher PGAn in WGD tumors compared with non-WGD tumors ($P = 3.3 \times 10^{-7}$), with HR−/HER2− tumors showing this pattern only in the Nigerian group (FDR = $3.4 \times 10^{-6}$; Supplementary Fig. 23). PGAn was also significantly correlated with mutation burden ($R = 0.55$, $7 \times 10^{-15}$; Supplementary Fig. 24).

We observed kataegis, a phenomenon of localized hypermutations often associated with genomic rearrangements[34], in 64.2% (111/173) of samples (Supplementary Data 2), with 3.6% of these, all Nigerian, harboring more than ten kataegis events (Supplementary Fig. 25). The majority of SNVs at kataegis foci were C > T and C > G mutations, associated with APOBEC mutational signatures SBS2 and SBS13. At the group level, the Nigerian group exhibited a higher number of foci (Supplementary Fig. 26) compared with the White (2.1-fold, $P = 6.4 \times 10^{-4}$) and Black groups (2.8-fold, $P = 0.002$);

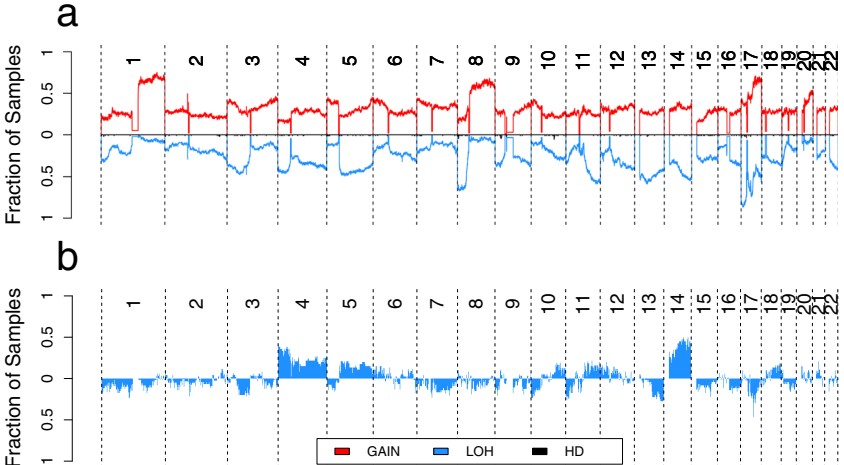

**Fig. 4 Homologous recombination deficiency analysis in breast cancer tumors across different ethnic groups.** Analysis of homologous recombination deficiency (HRD) in all three groups with samples ordered based on CHORD HRD type (i.e., proficient, *BRCA2*-type deficient, and *BRCA1*-type deficient). **a** Probability score of *BRCA1*- and *BRCA2*-type HRD and **b** prediction of HRD type (P(*BRCA1*) + P(*BRCA2*) > 0.5 cut-off for HRD). Unlabeled samples in **b** are considered homologous repair proficient. **c** Somatic mutations identified in homologous recombination repair pathway genes *BRCA1, BRCA2,* and *PALB2*. **d** Structural variant (SV) signature extraction identified seven signatures. S3 and S2 & S6 correspond to previously identified SV signatures correlated to *BRCA1*- and *BRCA2*-deficiency, respectively. These results were compared to HRD-associated **e** single-base substitution signature 3 (SBS3) and **f** insertion/deletion (INDEL) signatures 6 and 8 (ID6/ID8). **g** Clinical subtype of each sample.

**Fig. 5 Copy number landscape of Nigerian breast tumors. a** Genome-wide landscape of gain, loss of heterozygosity (LOH), and homozygous deletion (HD) events in the Nigerian group. The *y*-axis represents fraction of tumors with a particular event. LOH and HD are shown in opposite direction for better visualization. **b** Differential landscape (Nigerian versus White) of LOH events in HER2+ tumors. Events in the positive direction are more frequent in the Nigerian group. 14q LOH was virtually exclusive to the Nigerian HER2+ subtype. Source data are provided as a Source Data file.

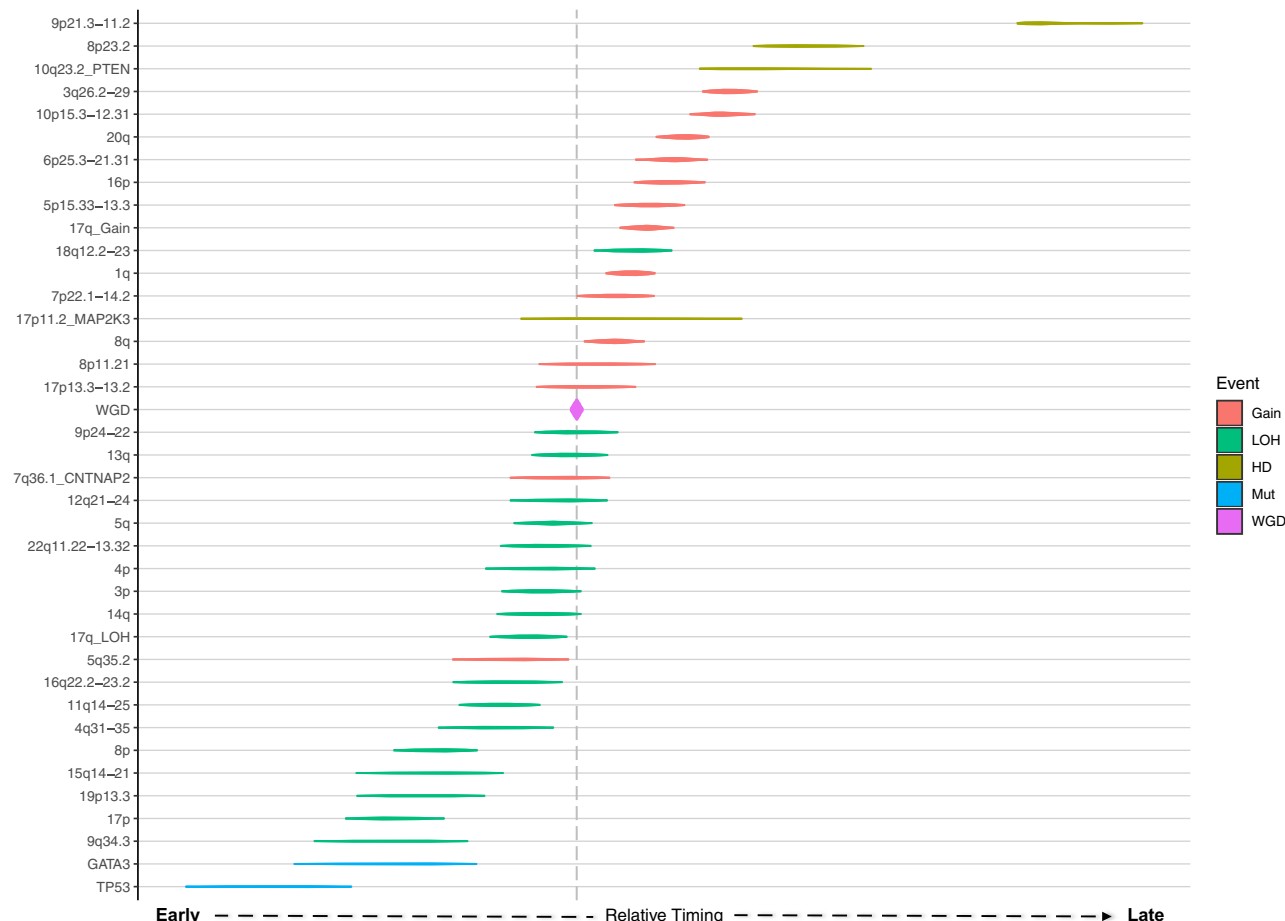

**Fig. 6 Chronological ordering of genomic events in Nigerian breast tumors.** Clonality-based ordering of significantly enriched copy number events (FDR < 0.05), whole-genome duplication (WGD), and key frequent mutational drivers (*TP53* and *GATA3*). A Plackett-Luce model was used to order the events by sampling from all possible tumor phylogenies across the entire dataset (1,000 iterations). Violins represent the 95% confidence interval of the relative timing estimate for each event. The events are ordered early to late by the mean value of the relative timing estimates. The vertical dotted line represents the mean timing estimate of WGD across all samples. LOH loss of heterozygosity, HD homozygous deletion, Mut mutational driver.

however, after adjusting for subtype, NRPCC and multiple-testing, it was only significantly higher than the White group (FDR = 0.017). No association was observed between the number of foci and subtype or age at diagnosis.

**Chronological ordering of genomic aberrations.** The Plackett-Luce probabilistic framework was used to reconstruct the most likely chronological order of genomic aberrations across all tumors. Genomic aberrations included in the analysis were enriched CNAs, WGD, and common mutational drivers, all of which were ordered based on clonality. Figure 6 depicts the relative timing of genomic events for the Nigerian dataset. Mutations in both *GATA3* and *TP53* were found to be early drivers. In addition to known early events such as 8p LOH and 17p LOH, 9q34.2 LOH, 14q LOH, 15q14-q21.3 LOH, and 19p13.3 LOH were among the early drivers. In contrast, in the White group, 19p13.3 (*STK11*) LOH did not occur pre-WGD (Supplementary Fig. 27), but 8p11.21 gain did occur pre-WGD. This gain event and 19p13.3 LOH encompass *ANK1* and *STK11* genes, respectively, both of which have been implicated in tumorigenesis[14,35], and copy number loss of *STK11* has been reported in metastatic breast cancer[36]. Ordering just the HR−/HER2+ subtype, 13q LOH, 14q LOH, and 8p11.21 gain occurred as early as the known early drivers (8p LOH and 17p

LOH), of which 14q LOH is virtually absent in the White HR−/HER2+ group.

**Higher intra-tumoral heterogeneity (ITH) in the Nigerian group.** ITH was assessed using weighted cancer cell fraction (wCCF), a metric that incorporates both the number and CCF of subclonal mutations. Significantly higher ITH was observed in cancers from the Nigerian than the White group (generalized linear model, $P = 0.005$; 3.4% increase) and Black ($P = 1.7 \times 10^{-4}$; 5.7% increase) groups after adjusting for clinical subtype and the higher sequencing depth of Nigerian samples. No significant difference was observed between White and Black groups ($P = 0.13$). The five samples with the highest subclonality (wCCF range 0.66–0.73) were all Nigerian. One of these samples had subclonal mutations in known driver genes *MAP2K4* and *RB1*[21], and a second sample in the previously unreported driver gene *F5*. The remaining three samples carried possible driver mutations in COSMIC tier 1 genes *ATR*, *ATRX*, and *KMT2D*, respectively. Further, mutations in *PRDM14*, an epigenetic regulator associated with an increase in ITH[37], were observed in two of the five samples (Supplementary Fig. 28).

**Identification of distinct genomic subtypes.** We analyzed the mutational and CNA drivers for potential pairwise interactions

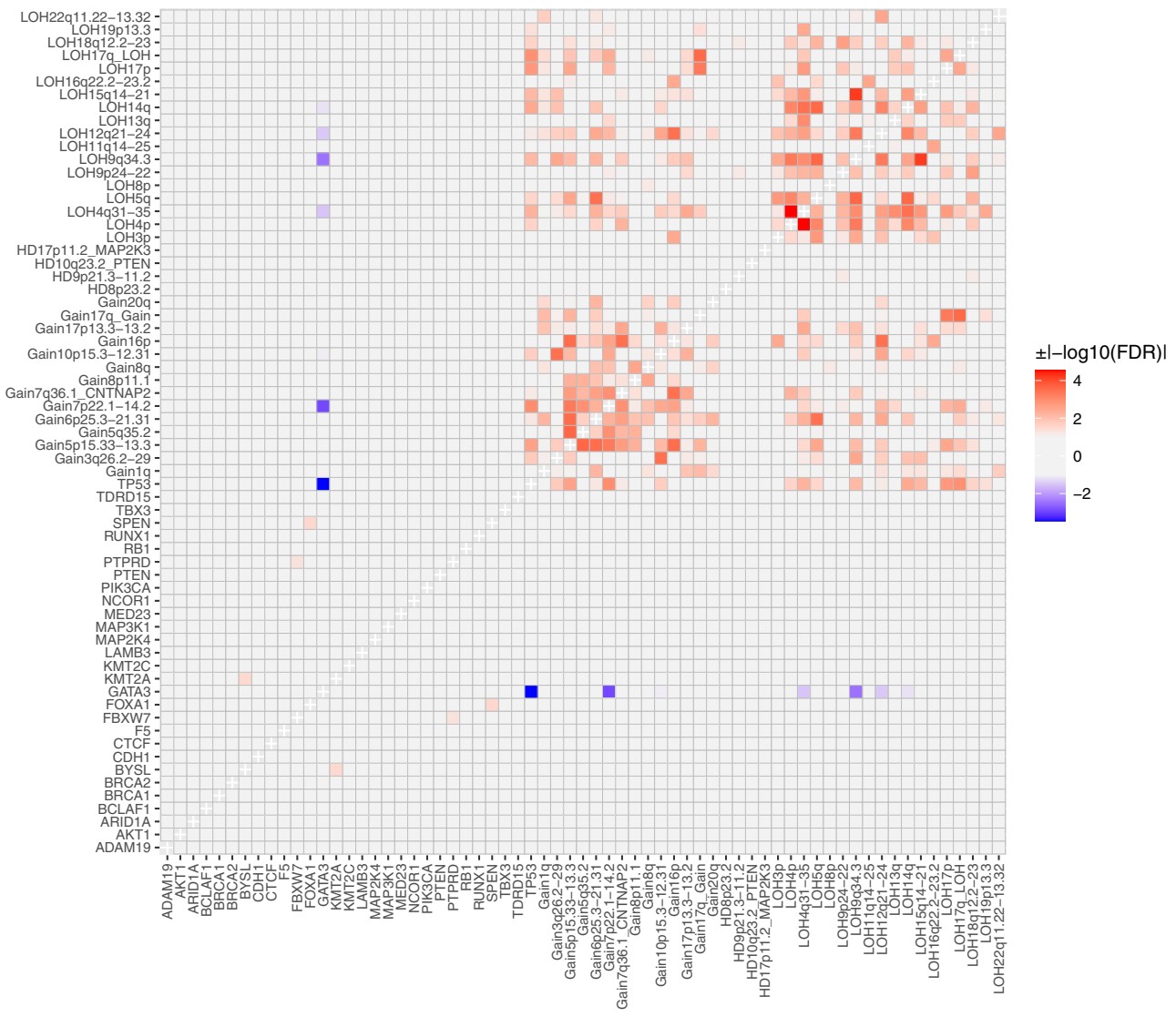

**Fig. 7 Somatic interaction analysis in breast cancer tumors.** Pairwise associations within and between mutational drivers and significantly enriched copy number aberrations were assessed using pairwise Fisher's exact test. Significant associations (two-sided P-values adjusted for multiple testing, FDR < 0.05) are shown with positive associations (co-occurrence; OR > 1) having a positive sign (red) and negative associations (mutual exclusivity; 0 < OR < 1) having a negative sign (blue).

and identified the majority to be co-occurrences of CNA events (Fig. 7). However, *GATA3* and *TP53* mutations were found to be almost mutually exclusive ($P = 2.46 \times 10^{-7}$, OR = 0.065, 95% CI 0.012–0.24) across groups and clinical subtypes (Supplementary Fig. 29). *TP53* and *GATA3* mutations had similar CCF distributions (Kolmogorov–Smirnov test, $P = 0.24$) and were predominantly clonal (95% and 96%, respectively). In addition, the timing model identified both genes as pre-WGD drivers. To assess likely gene function, we combined the CNA and mutation data to assess the rate of double-hits (biallelic inactivation) of these two genes. In total, 74 tumors had both clonal LOH and mutation at *TP53*. Half (49.5%) of these double-hit tumors had missense mutations; however, no difference was observed in their rate of WGD with those carrying loss-of-function mutations (OR = 1.04; $P = 1$), all *TP53* mutations were treated as equally deleterious in this context. WGD was significantly enriched in this double-hit subtype (FDR = $5.08 \times 10^{-5}$; Supplementary Fig. 30), but not the single-hit group. In contrast, of the 23 samples that carried clonal LOH at the *GATA3* locus, none had a *GATA3* mutation (Fisher's exact test, $P = 0.047$),

suggesting that either biallelic inactivation of *GATA3* is lethal for cells and therefore selected against or that loss of one *GATA3* copy causes haploinsufficiency. *GATA3* gene dosage was not associated with WGD.

The early, clonal, near mutually exclusive occurrence of *TP53* and *GATA3* suggests that they define distinct genomic subtypes of breast cancer, at least in Nigerian patients. We, therefore, proceeded to further characterize these subtypes in the Nigerian cohort. Interestingly, patients in the *GATA3* mutant subtype were diagnosed with an average of 10.5 years younger (42.9 versus 53.4 years, $P = 4.8 \times 10^{-4}$). From the chronological ordering of CNA events in each subtype (Supplementary Fig. 31), 5q35.1 gain was observed as an early event only in the *TP53* subtype. In contrast, HD at 9p21.3-p11.2 was an early event in the *GATA3* subtype, albeit with high variance due to the small number of samples with this event. The latter is an interesting observation given that HDs are reported to appear generally late in tumor evolution[38,39]. We observed no overall difference in the wCCF distribution of the two subtypes (Kolmogorov–Smirnov test, $P = 0.8$), suggesting no significant difference in ITH patterns.

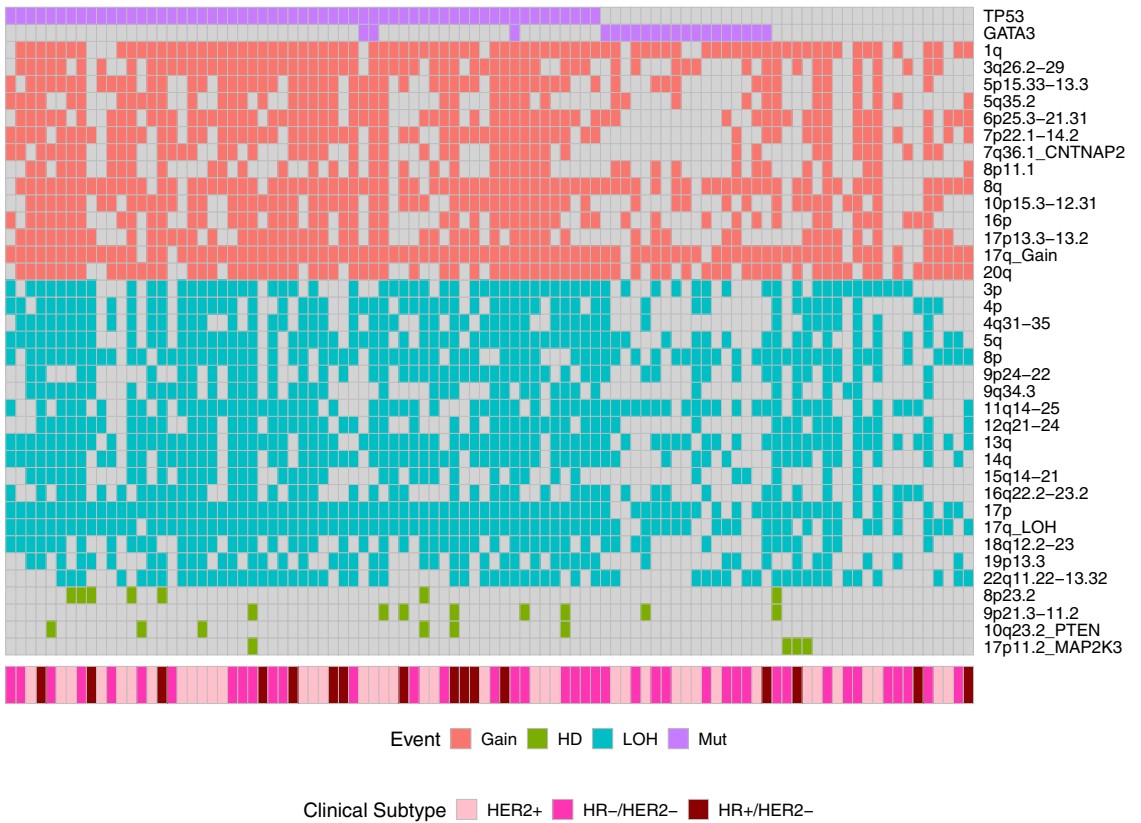

**Fig. 8 Copy number burden across genomic subtypes identified in the Nigerian group.** Distribution of significantly enriched copy number aberrations in the form of gain, loss of heterozygosity (LOH), and homozygous deletion (HD) were assessed for all Nigerian patients. *GATA3*-positive tumors displayed a lower copy number burden than *TP53*-positive tumors but were similar to tumors negative for both *TP53* and *GATA3*.

With respect to mutational signatures, we observed statistically significant increases in SBS1, SBS18, ID5, and the previously unreported INDEL-B in the *GATA3* subtype, while SBS8, SBS39, previously unreported DBS-D, ID8, and ID9 signatures were significantly over-represented in the *TP53* subtype (Supplementary Fig. 32). A higher prevalence of kataegis (OR = 6.79, $P = 0.057$) was observed in the *GATA3* subtype, which also affected more foci (1.53-fold, $P = 0.033$). In contrast, the *TP53* subtype showed greater GI in general with significantly higher mutation burden (1.84-fold, Wilcoxon test, $P = 0.007$) and WGD (OR = 7.28, $P = 0.004$), consistent with previous studies[40,41]. Mean PGA and PGAn were also both higher in the *TP53* subtype ($P = 7.6 \times 10^{-4}$ and $P = 0.012$).

In the Nigerian cohort, 20.6% of tumors ($n = 20$) were not mutated at either *TP53* or *GATA3* (Fig. 8). These samples had lower mutation burden ($P = 5 \times 10^{-4}$), WGD rate ($P = 0.007$), PGA ($P = 0.0013$) and PGAn ($P = 0.0014$) than the *TP53* subtype, and a lower rate ($P = 0.007$) and prevalence ($P = 0.02$) of kataegis foci compared with the *GATA3* subtype. Common clonal coding drivers were *PIK3CA* (25%; 28 and 5% in *TP53* and *GATA3* subtypes, respectively) and *RB1* (10%; 3 and 0% in *TP53* and *GATA3* subtypes, respectively) while 35% of these tumors had neither clonal mutations in known mutational drivers nor non-coding variants in either *ZNF217* or *SYPL1*. We therefore further explored this subset of 'quiet' genomes. We observed no difference in tumor purity between this subtype and the *GATA3* and *TP53* subtypes ($P > 0.24$; 0.42 versus 0.44 and 0.48, respectively), suggesting that the observation of lower GI in this subtype is not due to lower tumor content. Similar to the *GATA3* subtype, we observed an enrichment of INDEL-B compared with

the *TP53* subtype (FDR = 0.011) and a lower rate of enriched CNA events (Fig. 8). However, HD of 17p11.2 (*MAP2K3*) was an early event in the quiet tumors, present in 15% (3/20) of samples, while absent in the *GATA3* and late occurring in one tumor in the *TP53* subtypes respectively (Supplementary Fig. 31). Comparison of the whole-transcriptomes of a subset ($n = 49$) of samples from the three subtypes did not show a distinct cluster for the quiet genomes (Supplementary Fig. 33). However, this subtype demonstrated significant overexpression of genes (FDR < 0.05; Supplementary Fig. 34) previously associated with breast cancer including casein (*CSN1S1*, logFC = 7.7, $P_{adj} = 0.005$)[42] and Nectin-4 (*PRR4*, logFC = 4.1, $P_{adj} = 1.75 \times 10^{-8}$), Myomesin-2 (*MYOM2*, logFC = 2.77, $P_{adj} = 0.02$)[43], estrogen-related receptor beta (*ESRRB*, logFC = 2.51, $P_{adj} = 0.03$)[44] and neurotrophic receptor tyrosine kinase 3 (*NTRK3*, logFC = 2.45, $P_{adj} = 0.04$)[45], as well as genes associated with epithelial development (*SPRR2G/SPRR2E*, logFC ≥ 5.4, $P_{adj} < 0.02$), mucin production (*MUC7*, logFC = 4.7, $P_{adj} < 0.03$)[46], and metastatic potential (*LOXL4*, logFC = 2.63, $P_{adj} = 0.002$, and *SERPINE2*, logFC = 2.3, $P_{adj} = 0.006$)[47,48].

## Discussion
Previous genomic landscape studies have sketched out evolutionary trajectories of cancers including breast cancer, primarily focused on White patients of European ancestry ascertained in the US, Canada and Europe[12,14]. We used deep WGS to characterize the genomic landscape of somatic events and reconstruct the chronological ordering of events in breast tumors from 97 indigenous Nigerian women and compared the findings with tumors from White and Black patients in TCGA. We observed

key differences in somatic events occurring during the evolution of breast cancer in our Nigerian cohort, but these patients were not treated according to globally accepted clinical guidelines and lacked access to affordable cancer medicines. Nonetheless, we were able to gain improved understanding of breast cancer heterogeneity across populations.

First, genomic instability, a hallmark of cancer[49,50], was observed at a higher rate in the Nigerians in the form of WGD, PGA and kataegis, all of which may provide raw material for aggressive tumor behavior. Second, a higher level of ITH was observed in the Nigerian group. Given that ITH can serve as an indicator of tumor fitness for evolutionary adaptation and impinge upon the efficacy of therapeutic treatments[51], this may in part explain the biologically aggressive behavior of these tumors and poor clinical outcome in an unscreened population. Third, key somatic events were enriched in the HR−/HER2+ subtype, which may point to etiology but could also potentially impact response to HER2-targeted therapies. Of note, 14q LOH, which encompasses breast cancer genes such as *SERPINA1* and *DICER1*, was highly enriched in HR−/HER2+ Nigerian women. Loss of the former has been shown to be associated with poor outcome in this subtype[31] and that of the latter is associated with tumor progression and recurrence in this subtype[30]. Other enriched events include *LAMB3* (regulator of the PI3K/Akt signaling pathway in multiple cancers[52]) and the previously unreported INDEL-B (unknown etiology). Fourth, recent studies have demonstrated that WGS-based mutational signatures and HRD scores can reliably identify tumors with 'BRCAness' phenotype and these measures can be predictive biomarkers to guide treatment[53–55]. We were not only able to confidently stratify HR-deficient tumors by INDEL ID6/ID8 activity, CHORD HRD prediction and SV signatures, we were also able to delineate tumors into *BRCA1*- and *BRCA2*-types which fully correlate with *BRCA* driver mutations. Nigerian breast cancers with high frequencies of germline mutations[7] and over 30% HR-deficient tumors, therefore, constitute an identifiable 'BRCAness' population that could benefit from poly ADP-ribose polymerase inhibitors or platinum-based chemotherapy.

Lastly, epidemiological studies have shown a younger age of onset in women of African ancestry[56]. The significant enrichment of the early clonal driver *GATA3* in the Nigerian group and a positive trend in its recurrence with African ancestry (proportion trend test, $P = 0.0035$) along with a significantly lower age at diagnosis in patients with tumors carrying *GATA3* mutations is likely to be an underlying genomic event associated with young onset breast cancer. Furthermore, non-coding mutation hotspots at *ZNF217* and *SYPL1*, which are both associated with poor outcomes, and the previously unreported INDEL-B showed a strong positive trend with African ancestry, suggesting that these genomic features may also be associated with different evolutionary patterns of breast cancer in Nigeria.

Substantial progress has been made in unraveling the genomic complexity of breast cancer[12,22], one key improvement being the development of genomic classification of breast cancer[57,58]. In our cohort, we identified three genomic subtypes, of which the *GATA3* subtype was strongly enriched in the Nigerian group. These genomic subtypes presented distinct mutational properties. In the *TP53* subtype, all of which were double-hit *TP53* tumors, mutation burden was higher and WGD was significantly enriched. A recent study has shown that loss of both copies of *TP53* drives the poor outcome of patients with myelodysplastic syndromes, and different evolutionary trajectories were evident between single-hit and double-hit tumors[59]. In contrast, kataegis was more frequent in the *GATA3* subgroup. The quiet genome subtype displayed low genomic instability and demonstrated

associations to genes previously incorporated across breast cancer subtyping, prediction and prognostication approaches to date without a clearly consistent, previously described pattern. That a large proportion of the tumors had no known clonal driver and that a number of breast cancer related genes were highly upregulated suggests that tumor evolution in this subtype is complex and remains to be fully elucidated.

It is worth noting that our present study has a few limitations. We acknowledge the modest samples sizes reported here and that both TCGA and the present study are conducted on convenient and purposive samples ascertained in hospitals and may not reflect the origin populations. While starting to redress the imbalance with larger European cohorts, it will also supplement existing studies of tumor evolution in breast cancer. Nigerian and coastal West African populations contributed a significantly large proportion of genetic makeup of African Americans and African Caribbeans[60]. Compared with White women, Nigerian women with breast cancer have different epidemiological and genetic risk factor profiles, such as younger age at diagnosis, later age at menarche, higher parity, and a relatively high germline mutation rate in *BRCA1* and *BRCA2* genes[7,23]. Future studies integrating germline and somatic genetics, as well as epigenetic and environmental factors will extend our understanding of the dynamic nature of breast tumor evolutionary trajectories in African ancestry populations. Innovative science and technology when fully deployed can accelerate progress in tailoring screening for early detection to individuals at high risk. Complementary and agile liquid biopsy strategies[61–63] for known and de novo somatic alterations detection in oncogenic drivers (especially the early drivers) can be incorporated to immune surveillance strategies for prevention, early detection, and precision oncology care.

## Methods

**Patient cohort, ethics, and pathological assessment.** This study was embedded within the Nigerian Breast Cancer Study (NBCS) and approved by the Institutional Review Board of all participating institutions: The University of Chicago, University College Hospital, Ibadan (UCH), and Lagos State University Teaching Hospital (LASUTH)[1,7,9,64]. A grand total of 493 subjects were recruited from UCH (n = 284) and LASUTH (n = 209) between February 2013 and September 2015. Each patient gave written informed consent before participation in the study. Six biopsy cores and peripheral blood were collected from each patient. Two biopsy cores were used for routine formalin fixation for clinical diagnosis and the remaining four cores were preserved in PAXgene Tissue containers (Qiagen, CA) for subsequent genomic material extraction. In addition, 27 mastectomy tissues were preserved in RNAlater. Complete pathology assessment was performed centrally by study pathologists. Tumor burden was assessed based on cellularity, histology type, and morphological quality of tissue using TCGA best practices[57]. IHC on ER (rabbit monoclonal antibody, clone SP1 [Thermo Scientific, Cat# RM-9101]; 1:100 dilution), PR (rabbit monoclonal antibody, clone SP2 [Thermo Scientific, Cat# RM-9102]; 1:100 dilution), and HER2 (rabbit anti-human antibody, HercepTest Kit [Dako, Cat# K520421-5]; no dilution) were performed centrally in Nigeria and further reviewed in the United States. Cases with discordant results were again reviewed and resolved by the study pathologists. IHC scoring variables for Allred scoring algorithm were captured according to the 2013 ASCO/CAP standard reporting guidelines. Briefly, for ER and PR testing, immunoreactive tumor cells <1% was recorded as negative and those with ≥1% were reported positive. All the positive ER and PR cases were graded in percentages of stained cells and further scored in line with the Allred scoring system. Percentage of tumor staining for HER2 test were also reported along with a score of 0 and 1+ as negative, 2+ as equivocal, and 3+ as positive case. HER2 equivocal cases were further confirmed with genomic copy number calls.

**Sample selection and genomic material preparation.** Tumor samples containing >60% tumor cellularity were selected for DNA extraction using PAXgene Tissue DNA kit (Qiagen). Gentra Puregene Blood Kit (Qiagen) was used to extract genomic DNA from blood. Extracted DNA was quality controlled for its purity, quantity, and integrity. Identity of each extracted DNA sample was tested using AmpFlSTR Identifiler PCR Amplification Kit (Thermo Fisher Scientific). Samples that match >80% of the short tandem repeat profiles between tumor and germline DNA were considered authentic. RNA was extracted from PAXgene fixed tissues

using the PAXgene Tissue RNA kit (Qiagen). RNA integrity (RIN) was determined for all samples by the RIN score given by the TapeStation (Agilent) read out. RNA samples that had RIN scores of 4 and above were included in downstream sequencing analysis.

**Next-generation sequencing data generation**. A total of 100 WGS were performed at the University of Chicago High-throughput Genome Analysis Core (HGAC) and at the New York Genome Center (NYGC). Libraries were prepared using the Illumina Truseq DNA PCR-free Library Preparation Kit and were sequenced on an Illumina HiSeq 2000 sequencer at HGAC using $2 \times 100$ bp paired-end format and HiSeq X sequencer (v2.5 chemistry) at NYGC using $2 \times 150$ bp cycles. Mean coverage depth tumor was at $103.2\times$ and normal was at $35.1\times$. A total of 103 RNA-seq were carried out at the Novartis Next Generation Diagnostics facility. Average number of mapped reads per sample was 97 million. Seven samples failed QC and were excluded. Among the remaining 96 samples, 49 have WGS data available from the same patients. Total RNA were constructed into poly-A selected Illumina-compatible cDNA libraries using the Illumina TruSeq RNA Sample Prep kit. Passing cDNA libraries were combined in equimolar pools with other libraries of compatible adapter barcodes and later sequenced on the Illumina HiSeq 2500 sequencer.

**Alignment of DNA sequence to reference genome**. WGS reads were aligned to GRCh37 from GATK data bundle (v2.8; https://software.broadinstitute.org/gatk/) using BWA-MEM (v0.7.12; http://bio-bwa.sourceforge.net/). Duplicate reads were removed using PicardTools MarkDuplicates (v1.119; https://broadinstitute.github.io/picard/).

**Calling germline SNVs and indels**. Both SNVs and indels were called using Platypus[65] (v0.7.9.1; https://github.com/andyrimmer/Platypus) in single-sample mode. Only variants passing the Platypus 'PASS' filter were considered for downstream analysis.

**Calling somatic SNVs and indels**. SNVs were called using both MuTect[66] (v1.1.7; https://software.broadinstitute.org/cancer/cga/mutect) and Strelka[67] (v1.0.13; ftp://strelka:@ftp.illumina.com/v1-branch/v1.0.13/) with default parameters. Variants were called on the entirety of the genome in order to detect and retain any high-quality off-target calls. Any variant call that did not meet 'PASS' criteria for either algorithm was discarded. For a given tumor-normal pair, only SNVs called by both MuTect and Strelka were retained. Furthermore, we constructed a panel of 1,088 Nigerian and TCGA normal samples[8]. For a given normal sample, a site needed to be covered by a minimum of ten reads to be included. Any SNV that was supported by 5% or more of reads (MAPQ [MAPping Quality] ≥20; Base quality ≥20) in two or more samples was removed. SNVs were later annotated with Oncotator[68] (v1.5.3.0; https://software.broadinstitute.org/cancer/cga/oncotator) and those that met the required criteria ("COSMIC_n_overlapping_mutation >1" AND "1000gp3_AF ≤ 0.005" AND "ExAC_AF ≤ 0.005") were considered likely to be somatic and were retained. Small indels were called using cgpPindel (v3.0.1) within cgpWGS container (v2.0.1; https://dockstore.org/containers/quay.io/wtsicgp/dockstore-cgpwgs:2.0.1?tab=info) with default filters implemented. In addition, any indel calls found in the 1000 Genomes Project Phase 3 release[69] (http://ftp.1000genomes.ebi.ac.uk/vol1/ftp/release/20130502/ALL.wgs.phase3_shapeit2_mvncall_integrated_v5b.20130502.sites.vcf.gz) or the dbSNP[70] (b151; ftp://ftp.ncbi.nih.gov/snp/organisms/human_9606_b151_GRCh37p13/VCF/All_20180423.vcf.gz) were removed, unless they were found in the Catalog of Somatic Mutations in Cancer[71] (COSMIC v91; https://cancer.sanger.ac.uk/cosmic/download/CosmicCodingMuts.vcf.gz).

**Variant annotation**. SNVs and indels at both germline and somatic levels were annotated by ANNOVAR[72] (version May2018; http://annovar.openbioinformatics.org/) for functional consequence. In addition, variants were identified based on dbSNP (v150) and population frequency of variants were reported based on the Exome Aggregation Consortium dataset (ExAC v0.3; http://exac.broadinstitute.org/) and the Genome Aggregation Database (gnomAD v2.1.1; https://gnomad.broadinstitute.org/)[73,74].

**Clonality of somatic variants**. To measure clonality, we calculated cancer cell fraction (CCF) of each variant by adjusting the variant allele frequency (VAF) for copy number aberration (CNA) status, tumor purity and multiplicity of the variant[75]. Given that VAF of indels are reference-biased in standard variant calling algorithms, we used vafCorrect[76] (v5.7.0; https://github.com/cancerit/vafCorrect) to obtain accurate VAF from BAM files directly by leveraging unmapped reads. These re-estimated VAFs were used to calculate accurate CCF for coding indels. To assign coding mutations as clonal or subclonal, the CCF of all SNV and indel coding mutations were statistically assessed for clonal status. Briefly, the observed VAF was modeled using a binomial distribution and values representing the 95% interval were used to generate the 95% confidence interval (CI) of the observed CCF. Any variant with an upper CI above 1 was considered

to not deviate from a clonal state and, in turn, was assigned a CCF of 1. Otherwise, variants were considered subclonal and the original CCF value was retained. This allowed us to assess the clonality of coding mutations without introducing an arbitrary CCF cut-off.

**Somatic drivers**. cDriver[17] (v0.4.2; https://github.com/hanasusak/cDriver) was used to identify cancer drivers by not only taking into account the recurrence against the background mutation rate and functional impact (CADD score v1.6; https://cadd.gs.washington.edu/)[77], but also the CCF of each variant. In addition, MutSigCV[18] (v1.3; https://software.broadinstitute.org/cancer/cga/mutsig) was used independently to identify drivers based on recurrence given background mutation processes. The 20/20 principle[19] was applied to all detected drivers to classify, based on mutation patterns in the dataset, which are tumor suppressor gene (TSG), oncogene (ONC) or both. Enrichment analysis was undertaken for all mutational drivers (detected and previously known; n = 30) across ethnic groups, clinical subtypes and ER status using Fisher's exact test to identify differential prevalence of drivers. The mutational landscape plot was generated using Maftools[78] (v2.6.05; https://www.bioconductor.org/packages/release/bioc/html/maftools.html).

**Non-coding hotspots**. We partitioned each chromosome in the genome into discrete bins of 100 kb and undertook a genome-wide screening of variant recurrence in each of the non-overlapping bins. Similar to a genome-wide association study construct, we compared the Nigerian group with both the White and Black groups using pairwise Fisher's exact test followed by multiple testing correction for differential prevalence to detect potential non-coding mutation hotspots in or near coding genes enriched in the Nigerians. This analysis was based only on SNVs since overall rate of SNV was not significantly different between Nigerian and the other two groups. Therefore, no overall bias is present in the rate of SNVs and local over-representation signals are likely to be genuine ethnicity-specific hotspot signals.

**Mutational signatures**. De novo extraction and decomposition to known cosmic mutation signatures in single-base substitution (SBS), double-base substitution (DBS), as well as small insertion and deletion (ID) formats were implemented based on a non-negative matrix factorization (NNMF) framework using SigProfilerExtractor[28] (v0.0.5.77; https://github.com/AlexandrovLab/SigProfilerExtractor). Because NNMF is more accurate with a larger number of samples[79], we increased our sample set by adding 128 additional breast cancer WGS samples from the Pan-cancer Analysis of Whole Genomes (PCAWG) study[80] and eight TCGA (Asian and unassigned ancestries) WGS samples[8]. Our final input dataset for SigProfilerExtractor thus included a total of 309 samples. Signatures identified as singletons in the entire dataset were removed from analysis.

**Structural variant (SV) calling**. Three different algorithms were used to identify structural variants in BAM files aligned to the hg19 human genome: Manta (v1.1.0; https://github.com/Illumina/manta)[81], DELLY (v0.7.0; https://github.com/dellytools/delly)[82] and the lumpy-express function of Lumpy (v0.2.13; https://github.com/arq5x/lumpy-sv)[83]. Calls from blacklisted regions (http://cf.10xgenomics.com/supp/genome/hg19/sv_blacklist.bed) and segmental duplication regions (http://cf.10xgenomics.com/supp/genome/hg19/segdups.bedpe) were filtered out using SURVIVOR (v1.0.6; https://github.com/fritzsedlazeck/SURVIVOR)[84]. The SVs from all three algorithms were then merged with SURVIVOR, using a 2-vote consensus approach to generate the final set of variants, ensuring an improved precision on the SVs called[85].

**Homologous recombination deficiency (HRD) prediction**. Prediction of HRD from somatic mutations was performed with the machine-learning model CHORD (v2.0; https://github.com/UMCUGenetics/CHORD)[29]. Generated VCF files for SNVs, indels and SVs were provided as input to the classifier. HR-deficient samples were labeled as either *BRCA1*-like, *BRCA2*-like or unknown.

**SV signatures**. Breast cancer SV signatures were obtained using signature.tools.lib[86]. For the SV signatures extraction, ten bootstraps per catalog were used in conjunction with brunet non-negative matrix factorization (NMF)[87], and clustering with matching algorithms. Clustering with matching prevents signatures from the same NMF run being assigned to the same cluster[86]. To determine the actual number of SV signatures within our dataset, rearrangement catalogs were clustered multiple times, using different cluster numbers. Each attempt was evaluated based on the clustering average silhouette width and the average difference between the reconstructed signature catalogs and the original catalogs, resulting in seven SV signatures. Six out of the seven signatures extracted are highly comparable to the rearrangement signatures reported by Nik-Zainal et al.[22]. Signature 6 (S6) is likely to be a previously unreported signature in our African-centric dataset that has not been identified previously. A minimum of 10% contribution was used to call a signature active in a sample.

**Calling somatic CNA**. Genome-wide copy number profiles of all samples in the entire dataset were obtained by Battenberg (v2.2.8; https://github.com/Wedge-lab/battenberg)[22,39]. In addition to calling clonal and subclonal allele-specific CNAs, it was also used to estimate purity and average ploidy of each tumor. As part of quality control analyses, in the entire dataset, we detected three samples in the Nigerian group which had very low purity estimates based on copy number analysis (<10%). However, one sample did not show a consistent low mutation burden ($n_{SNV}$ = 4,285, $n_{SNV}$ of other two samples <100). An SNV-centric VAF-based purity estimation analysis was undertaken to independently evaluate the purity estimate for this sample. Briefly, all SNVs within diploid regions were identified and VAFs were calculated. In the VAF density distribution, the local peak with maximum VAF was considered as the clonal peak. The purity, calculated as 2*clonalVAF, was consistent with that based on the CNA analysis. These samples were thus removed from subsequent analyses in the Nigerian cohort (final $n = 97$). To call recurrent CNA events, first, CNA of each type (i.e. Gain, LOH and HD) were aggregated across all samples along the chromosomes to obtain the frequency landscape of each CNA type based on all observed breakpoints. Next, a permutation test ($n = 1,000$) followed by multiple testing correction was undertaken to identify regions that were significantly enriched above the random background copy change rate. The enriched regions that encompassed the HLA region (6p21), or specific to telomeric ends or present as a singleton were excluded.

**Genomic instability analysis**. Whole-genome duplication (WGD) was called in samples where the proportion of the genome with balanced 2:2 copy number status was larger than that with 1:1 diploid copy number. Samples were also manually inspected to see WGD features such as multiple copy losses post-WGD (3:1 copy number status) and LOH events with 2:0 status. For the reconstruction of the chronological ordering of somatic events, WGD, as an event, was called in samples that had an average ploidy greater than three[39]. Proportion of genome altered (PGA), which is the proportion of genome bases encompassed by CNAs, was calculated for all samples based on the Battenberg output. In WGD samples, PGA was calculated as the proportion of genome that did not have a balanced tetraploid copy number state (i.e. 2:2). PGA does not take into account the number of CNA events. A modified metric (PGAn) was calculated as the geometric mean of PGA and number of breakpoints to not only take into account length of CNA segments, but also the number of CNA segments to allow for genomes with focal or global shattering. We followed previous studies in defining kataegis events[22,79]. KataegisPCF (v1.0; https://github.com/nansari-pour/KataegisPCF) was used to detect the kataegis loci and visualize the kataegis events based on SNVs. A minimum of six consecutive SNVs with mean distance ≤1 kb were required for kataegis events, which were identified systematically by applying piecewise constant fitting (PCF)[88] on inter-variant distance of all SNVs across the genome.

**Timing model of ordering events**. To reconstruct the chronological ordering of somatic events, we developed a timing model to order the occurrence of mutational drivers and enriched CNAs based on the clonality of the events. Briefly, for CNAs, Battenberg copy number calls were used to assign clonality (whether CCF = 1 or <1) and describe their type (i.e. gain, LOH and HD). CCF of each variant was estimated by adjusting VAF according to the CNA status of the locus and purity of the tumor sample[75]. Variants were then classified as clonal (CCF = 1) and sub-clonal (CCF < 1). All events were combined per sample and ordered based on CCF. Where more than one tree could be inferred based on subclonal events, all possible trees were generated and randomly chosen in each iteration of ordering events. To time the events based on the entire dataset, events were ordered based on clonality (randomized clonal events followed by a sampled tree of subclonal events) in each sample. To classify events with respect to WGD, we used major/minor copy number status and the estimated number of chromosomes bearing the mutation (NCBM) to call pre-WGD and post-WGD CNA and mutations respectively by using logical rules for CNA[39] and extended them here for mutations. For instance, in a tumor with WGD, a clonal coding mutation with NCBM ≥ 2 was considered as a pre-WGD event while that with NCBM = 1 was defined as post-WGD. The Plackett-Luce model[89,90] for ordering partial rankings was implemented using the PlackettLuce package in R (v0.3.0; https://github.com/hturner/PlackettLuce) based on the ordering matrix of the entire dataset to infer the order of events at the population level while allowing for unobserved events in individual tumors. This analysis was undertaken for 1,000 iterations to obtain the 95% CI of the timing estimate of each event. In this implementation of the Plackett-Luce model, the clonality level of an event across the population dictates the overall ranking. However, its frequency affects the variance of the timing estimate, such as rarer events show higher 95% CI. We repeated this analysis within each clinical and genomic subtype.

**ITH analysis**. To infer subclonal architecture of each tumor, a Bayesian Dirichlet process algorithm was implemented (DPClust v2.2.2; https://github.com/Wedge-lab/dpclust), to cluster somatic SNVs based on CCF[75,91]. Mutation clusters were identified as local peaks in the posterior mutation density obtained from DPClust. In addition to the clonal cluster, the number of subclonal clusters and their respective mutation burden were also estimated. To quantify subclonality, we

calculated weighted CCF (wCCF) which is defined by the mean of the CCF of mutation cluster peaks adjusted by the mutation burden of clusters. The ability to detect subclones depends, not on the number of detected SNVs, but on the number of reads per chromosome copy (NRPCC)[92]. This metric takes tumor purity, ploidy and sequencing coverage into account. We control for this effect by including only tumors with NRPCC ≥ 10. In these tumors, we should be sufficiently powered to detect subclones with CCF > 0.3.

**Somatic interactions**. To test for somatic interactions, we undertook mutual exclusivity and co-occurrence analysis by using pairwise Fisher's exact test to detect significant pairing within and among mutational driver and CNA events. Negative associations with an odds ratio (OR) between 0 and 1 (exclusive) were considered mutually exclusive and positive associations with OR > 1 were considered co-occurring with the magnitude of OR being inversely and directly proportional to the strength of the association respectively.

**Estimating genetic ancestry of study population**. DNA samples from blood or normal breast were genotyped using Affymetrix SNP 6.0 arrays. For patients with both blood and normal breast samples, we utilized the genotype data from blood only. Uncorrelated single nucleotide polymorphisms from the TCGA cohort and the International HapMap Project were included in the principal component analysis. The top two eigenvectors from principal component analysis were plotted and the three known continental ancestry groups from the HapMap were used as anchors. The proportion of ancestry relative to the reference continental groups for each patient was estimated by projecting the eigenvectors onto each of the three axes defined by the three anchors. The genetic ancestry information of breast cancer patients from TCGA was obtained from our previous study[6]. Briefly, we estimated the ancestry of breast cancer patients from TCGA using principal component analysis. According to the estimated proportion of ancestry, patients were grouped into genomic Black (≥50% African ancestry), genomic White (≥90% European ancestry), and genomic Asian (≥90% Asian ancestry). All Nigerian patients were assumed to be 100% African with little to no admixture with other populations.

**RNA-seq and differential gene expression analysis**. Read alignment of RNA-seq to GRCh37 (hg19) as reference genome and GENCODE[93] (v19; https://www.gencodegenes.org/human/release_19.html) for gene annotation was performed using STAR[94] (v2.4.2a; https://github.com/alexdobin/STAR) and HTSeq[95] (v0.6.1p1; https://github.com/htseq/htseq). Quality control metrics were calculated using RNA-SeQC[96] (v1.1.8; https://software.broadinstitute.org/cancer/cga/rna-seqc), featureCounts[97] (v1.5.1; http://subread.sourceforge.net/), PicardTools (v1.128; https://broadinstitute.github.io/picard/), and SAMtools[98] (v1.3.1; http://www.htslib.org/). Differential expression analysis of raw read counts of protein-coding genes from HTSeq was then performed using DESeq2[99] (v1.24.0; https://bioconductor.org/packages/release/bioc/html/DESeq2.html), with subtype information based on immunohistochemistry to maintain consistency with genomic data. Analysis was performed with ancestral populations only to avoid batch effect artifacts. PAM50 molecular subtyping was performed[100].

**Statistical methods**. All statistical calculations were implemented in R (v3.4.3; https://www.r-project.org/). For categorical data, we used Fisher's exact test (*sher.test*) and for continuous data, we used wilcoxon rank test (*wilcox.test*) or Student's t-test (*t.test*) wherever appropriate. Where applicable, P-values were adjusted for multiple testing (*p.adjust*) based on the false discovery rate (FDR) proposed by Benjamini and Hochberg[101] with FDR < 0.05 considered significant, unless stated otherwise. This was done to not only reduce type I error, but to also minimize type II error[102].

Given the difference in coverage between Nigerian and TCGA WGS samples, to detect true differences in ITH, a generalized linear model (*glm*) was used to model the association of the ITH metric (wCCF) with ethnicity while adjusting for the confounding effect of the covariable NRPCC. The Cochran-Armitage trend test (*prop.trend.test*) was used to assess whether proportions of a variable across the three groups were monotonic with the ordered variable (i.e. increasing African ancestry proportion). The two-sample Kolmogorov–Smirnov test (*ks.test*) was implemented to detect significant differences in the distribution of a variable across different groups.

**Reporting summary**. Further information on research design is available in the Nature Research Reporting Summary linked to this article.

## Data availability

The raw sequencing data and the processed genomic data from Nigerian cases have been deposited in dbGaP under Study Accession phs001687.v1.p1. TCGA raw sequencing data are available in dbGaP under Study Accession phs000178.v11.p8. Data access to dbGaP can be obtained by contacting National Cancer Institute Data Access Committee (NCIDAC@mail.nih.gov). Access to TCGA variant calls that support the findings of this study are available on request to the corresponding author (O.I.O) from the requestor who has approved authorized access to TCGA controlled data. The remaining data are

available within the Article and Supplementary Information. Source data are provided with this paper.

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

## Acknowledgements

We are greatly indebted to all the patients who agreed to participate in this study and graciously donated their biological materials. This study was supported by National Institutes of Health (U01 CA161032, R01 MD013452, P20-CA233307), Susan G. Komen for the Cure (SAC110026, SAC210203) and Breast Cancer Research Foundation (BCRF-19-120) to O.I.O.. O.I.O. is an American Cancer Society professor. A.J.G. was funded by a postdoctoral research fellowship (grant number: P2BSP3_178591). P.V.L. was supported by the Francis Crick Institute, which receives its core funding from Cancer Research UK (FC001202), the UK Medical Research Council (FC001202), and the Wellcome Trust (FC001202). P.V.L. is a Winton Group Leader in recognition of the Winton Charitable Foundation's support towards the establishment of The Francis Crick Institute. The computational aspects of this research were also funded from the NIHR Oxford BRC with additional support from the Wellcome Trust Core Award (203141/Z/16/Z). D.H. was also supported by National Institutes of Health (R01-CA228198) and Breast Cancer Research Foundation (BCRF-20-071). We want to thank the New York Genome Center and Novartis Institutes for BioMedical Research (NIBR) for the quality of the sequencing services provided. Special thanks to the West Africa Breast Cancer Study investigators for their contributions to the project. We also want to thank Qiagen for their generous donation of PAXgene Tissue Containers and DNA Extraction Kits for this study. The views expressed are those of the author(s) and not necessarily those of the NHS, the NIHR or the Department of Health.

## Author contributions

O.I.O., D.H., O.O., and J.O. co-conceived the study. N.A.-P., Y.Z., and T.F.Y designed the experimental approach. N.A.-P. led the computational analyses, undertook statistical analyses and interpreted results. Y.Z., T.F.Y., J-B.R, A.T, J.J.P., S.D., A.W., P.S.R., and A.J.G. conducted bioinformatics analyses and interpreted results. D.F. provided computational support. A.P., Dr. A.G.F., C.P.B., T.O., and N.I. recruited the patients and collected specimens from patients. A.S. and M.A. performed pathological assessment of patient specimens. A.O. procured patient specimens and prepared DNA/RNA. O.O. served as Site-PI and provided overall supervision of the study at UCH. J.O. served as Site-PI and provided overall supervision of the study at LASUTH. J.B. served as Site-PI and provided overall supervision of the study at NIBR. M.C. and D.H. provided statistical support and discussion. P.V.L. and K.P.W. provided supervisory support and discussion. N.A.-P., Y.Z., and P.S.R. drafted the manuscript. All authors reviewed and edited the manuscript. O.I.O. and D.C.W. equally supervised this work.

## Competing interests

K.P.W. is a Scientific Advisor and Fellow at Tempus. O.I.O is co-founder at CancerIQ, serves as Scientific Advisor at Tempus and is on the Board of 54gene. All other authors declare no competing interest.

**Additional information**

[1]Big Data Institute, Nuffield Department of Medicine, University of Oxford, Oxford OX3 7LF, UK. [2]MRC Molecular Haematology Unit, Weatherall Institute of Molecular Medicine, University of Oxford, Oxford, UK. [3]Center for Clinical Cancer Genetics and Global Health, Department of Medicine, The University of Chicago, Chicago, IL 60637, USA. [4]Department of Pathology and Forensic Medicine, Lagos State University Teaching Hospital, Ikeja, Lagos, Nigeria. [5]Department of Pathology, University of Ibadan, Ibadan, Oyo, Nigeria. [6]Manchester Cancer Research Centre, University of Manchester, Manchester M20 4GJ, UK. [7]Cancer Science Institute of Singapore, National University of Singapore, Singapore 117599, Singapore. [8]European Molecular Biology Laboratory, European Bioinformatics Institute, Cambridge CB10 1SD, UK. [9]Wellcome Trust Sanger Institute, Hinxton, Cambridge CB10 1SA, UK. [10]Department of Computer Science, The University of Chicago, Chicago, IL 60637, USA. [11]Institute for Genomics and Systems Biology, University of Chicago, Chicago, IL 60637, USA. [12]Institute for Advanced Medical Research and Training, College of Medicine, University of Ibadan, Ibadan, Oyo, Nigeria. [13]Oncology Unit, Department of Radiology, Lagos State University, Ikeja, Lagos, Nigeria. [14]Department of Pharmaceutical Chemistry, Faculty of Pharmacy, University of Ibadan, Ibadan, Oyo, Nigeria. [15]Department of Surgery, University College Hospital, Ibadan, Oyo, Nigeria. [16]Department of Surgery, Lagos State University Teaching Hospital, Ikeja, Lagos, Nigeria. [17]Girona Biomedical Research Institute (IDIBGI), Hospital Universitari de Girona Dr Josep Trueta, Girona, Spain. [18]The Francis Crick Institute, London NW1 1AT, UK. [19]Department of Human Genetics, The University of Chicago, Chicago, IL 60637, USA. [20]Section of Genetic Medicine, Department of Medicine, The University of Chicago, Chicago, IL 60637, USA. [21]Tempus Labs Inc., Chicago, IL 60654, USA. [22]Centre for Population and Reproductive Health, College of Medicine, University of Ibadan, Ibadan, Oyo, Nigeria. [23]Department of Public Health Sciences, The University of Chicago, Chicago, IL 60637, USA. [24]These authors contributed equally: Naser Ansari-Pour, Yonglan Zheng. ✉email: david.wedge@manchester.ac.uk; folopade@medicine.bsd.uchicago.edu

