## [Peer Review File · Nature Communications]

Whole-genome Analysis of Nigerian Patients with Breast Cancer Reveals Ethnic-driven Somatic Evolution and Distinct Genomic SubtypesReviewers' Comments:

Reviewer #1:

Remarks to the Author:

In the manuscript titled "Whole-genome Analysis of Nigerian Patients with Breast Cancer Reveals Ethnic-driven Somatic Evolution and Distinct Genomic Subtypes", the authors used deep WGS to characterize the genomic landscape of somatic events and reconstruct the chronological ordering of events in breast tumors from 97 indigenous Nigerian women and compared the findings with tumors from White and Black patients in TCGA.

Major comments:

1. Genomics in another racial/ethnic background (e.g. indigenous Nigerian women) did not provide enough innovation discovery, practicability in breast cancer research, or any targeted therapeutic application to this race or a novel target.
2. The author should do some, or at least one following experiment to prove their analysis results: a) biological significance of mentioned thirteen driver genes. b) the function of mentioned genes in 14q. c) relationship between key somatic events in the HR-/HER2+ subtype and their impact on HER2-targeted therapies d) the biological influence of enriched events including LAMB3, GATA3, or non-coding mutation hotspots at ZNF217 and SYPL1. e) providing any actionable target and the effect of its inhibitor

Minor comments:

1. What about the variant allele frequencies of the top mutated gene? Any useful comparison with the two other cohorts?
2. Full name of WGD should be first shown in page 10.
3. What about characteristics of mutations in oncogenic signaling pathways?
4. Though GATA3 and TP53 mutations were found to be almost mutually exclusive, how about the distribution among cohorts or molecular subtypes.
5. It would be better to use bar plot or pie plot to present information about molecular subtypes between cohorts

Reviewer #2:

Remarks to the Author:

Major concerns.

First, there seems to be a bias in all three cohorts. In all three cohort only minority of included cases suffers from HR+/HER2neg disease while this in general is the most common subtype both in Caucasians as well as in African people. For now the bias in the cohorts could be an explanation of the differences observed. Also the TCGA reference cohort is small. If the author would have included 560 ICGC genomes (mostly caucasian, if I am not mistaken) the reference cohort would be more representative and the power of the study much larger. In any case the bias requires an explanation. The Nigerian cohort was 40% HER2 and other African cohort had zero. Are these representative percentage for these populations?

Second, if there is not a selection bias, another explanation would be that negative ER status reported was false negative in certain percentage of cases. This could explain why GATA3 mutants are seen among the ER negative cases and while in fact in general them being part of the ER positive cases. The fact that GATA3 is frequently inactivated in triple negative breast cancer (TNBC) is novel and interesting but also unexpected and remarkable. GATA3 is not expressed in the basal layer of the mammary epithelium if I am not mistaken. Inactivation of GATA3 in that cell type will thus not drive tumorigenesis. Thus, if true, the proposed GATA3 inactivation is occurring in a luminal cell type causing it to develop in an ER negative tumor which is possible. This observation, however, demands for independent confirmation and using an orthogonal method. Staining tumors and appropriate

controls may help to support the finding but does not fully falsify the finding. I am also interesting in the type of GATA3 mutations observed in the TNBC of this cohort (data not excluded in the submitted manuscript or I overlooked them). In luminal BC, and reported by many groups, the GATA3 mutations are always out of frame mutations and occur in the latter 2 exons) and occur in a specific domain (last zinc finger) of the protein and occur on one allele. This is more in line with a dominant phenotype than the calculated tumor suppressor role here. Tumor suppressor mutations in a gene are wide spread and not localized. The GATA3 mutations in these TNBC cases are expected to be different at least else I find it hard to explain why they occur in TNBC.

Third. Mutual exclusivity between TP53 and GATA3 is reported. This can easily be explained by them being present in a different subtype. This is what is seen in causation and all reported cohorts thus far. But if indeed present in TNBC one might argue that they occur in different TNBC subtypes. Since there is RNAseq for a subset could this shed light on this.

Fourth, signatures are called using non-negative factorization (nnF) which makes sense and using the newest algorithm of Alexandrov. However, using limited sized cohorts there is a risk of misclassification. Did the author also pursue de novo calling of signatures. This does not force mutations in upfront defined classes and might provide independent confirmation if SBS39 is correctly called or not. Some is true for other novel signatures. Including larger cohorts could help here as well.

Fifth, many differences (drivers, CNAs, heterogeneity, etc) between groups (black (Nigerian or other) and white) are not corrected for subtype. One cohorts (non-Nigerian Africans) lack even a subtype. This precludes proper comparison particularly in the subgroup being absent. It does not become clear how this was corrected for (apart from the bias).

HRD is defined based on signature 3 only. This is a poor classifier of HRD. Using additional information such as indel size and number, certain rearrangements (tandem duplications) etc as defined by Nik-Zainal and coworkers (HRDirect) or Cuppen E and coworkers (Chord) are superior. Why were these not used to classify HRD.

I wonder whether Nigerian associated CNAs (e.g. 14q LOH) hold if larger control cohorts are included which are widely available.

Also more kataegis spots are observed in Nigerian cases. However, the Nigerian cohort is HER2 enriched and APOBEC mutagenesis high in HER2 cases. So will this hold in better matched groups or when corrected for subtype.

I am not sure why the comparisons was not also compared to the ICGC cohort involving 560 breast cancers (Zainal et al). Particularly the significant findings. I am not sure whether ancestry is reported for all cases in that cohort but it will increase power to find significant differences.

I would not use PGA since it is generally known to be inferior to the number of aberration in a genome when calling genomic instability as it ignores events being often independent.

Minor points.

I would not use black and white but African versus Caucasian. It is also not defined where the non-Nigerian Africans originate from. This should be made more clear. Are the African Americans or non-Nigerian Africans.

HER2 cases in the Nigerian cases was high. Was this also technical are was this confirmed in the WGS data.

John Martens

Was the sequence depth between cohort the same and the sequencing of the same quality. This might

affect for instance the number of Karyotypic foci called as well as the clonality scores.

RNA data are not provided so they can be deleted from the manuscript. Unless may be used to independently confirm phenotype.

Reviewer #3:

Remarks to the Author:

This is a very interesting analysis of somatic tumor genetics in a understudied population. Specific comments are below:

There was a lot of work investigating HRD but some of this could be improved noting the below points

1) Did the authors run a rearrangement signature analysis similar to determine if BRCA1/2 are correlated with the signatures in this paper
<https://www.nature.com/articles/nature17676>

2) Similarly, with the BRCA1/2 HRD signature are they BRCA1-like or BRCA2-like ?
<https://www.nature.com/articles/s41467-020-19406-4>

or HRDetect (<https://www.nature.com/articles/nm.4292>) from
A. Degasperi et al. A practical framework and online tool for mutational signature analyses show intertissue variation and driver dependencies, Nature Cancer, [<https://doi.org/10.1038/s43018-020-0027-5>], 2020.

3) DO the authors know the biallelic status of BRCA1/2 and if biallelic have a HRD signature but not monoallelic?

4) Can the authors clarify if the hotspot non-coding analysis is somatic or germline? I suspect somatic but maybe useful to explain in the legend of Figure 2 as they note variants. In general, the legend will benefit from more details to make it clearer to readers

5) ZNF217 non-coding hotspot found in 40% of AA compare to 4% in whites? are there confidence intervals for this estimate and does gene expression data in TCGA support non-coding mutations are related to altered levels of expression in AA?

6) The samples used from the study are a subset of the larger Nigerian study population how representative are these samples from the overall study in terms of risk factors/tumor characteristics?

Jonine Figueroa

Response to reviewers

Ansari-Pour N & Zheng Y *et al.* - NCOMMS-20-49388-T

REVIEWER COMMENTS

Reviewer #1, expert in breast cancer genomics and mutational signatures (Remarks to the Author):

In the manuscript titled "Whole-genome Analysis of Nigerian Patients with Breast Cancer Reveals Ethnic-driven Somatic Evolution and Distinct Genomic Subtypes", the authors used deep WGS to characterize the genomic landscape of somatic events and reconstruct the chronological ordering of events in breast tumors from 97 indigenous Nigerian women and compared the findings with tumors from White and Black patients in TCGA.

Major comments:

1. Genomics in another racial/ethnic background (e.g. indigenous Nigerian women) did not provide enough innovation discovery, practicability in breast cancer research, or any targeted therapeutic application to this race or a novel target.

We agree with this broad statement because it is never possible to achieve all the steps described by this reviewer in one manuscript. We apologize that we did not sufficiently discuss the importance of genomic studies of populations of non-European Ancestry to Precision Oncology to satisfy this reviewer. Moreover, the goal of the study was not to describe the mutational landscape or to seek druggable targets, but this study represents the largest WGS-based breast cancer Life History study on patients of African ancestry to date. The diversity in breast cancer genomes is only now being discovered and the novel findings from this study pave the way for future focused genomic studies in non-European populations that have traditionally been excluded because of entrenched structural barriers in the biomedical research enterprise.

2. The author should do some, or at least one following experiment to prove their analysis results: a) biological significance of mentioned thirteen driver genes. b) the function of mentioned genes in 14q. c) relationship between key somatic events in the HR-/HER2+ subtype and their impact on HER2-targeted therapies d) the biological influence of enriched events including LAMB3, GATA3, or non-coding mutation hotspots at ZNF217 and SYPL1. e) providing any actionable target and the effect of its inhibitor

While we agree that functional validation of the 'candidate drivers' is warranted, the additional experiments requested are beyond the scope of this report. We acknowledge that a robust framework is required to precisely characterize a gene as a driver (tumor suppressor gene [TSG] or oncogene [ONC]). Currently, the state-of-the-art system has been developed by COSMIC (Catalogue of Somatic Mutations in Cancer), which is a community joint-effort to describe and curate the comprehensive cancer genes based on multiple lines of evidence. We also admit that much work needs to be done to better distinguish TSG/ONC within this context, since many factors have impact on that, including genetic alterations (not just the gene itself), tissue of origin

(and subtypes), tumor stage, environmental factors, and tumor microenvironmental factors. Such depth of analysis is beyond the scope of the current study, and it can result in a series of publications of its own. In this study, we identified 13 ‘candidate driver’ genes which comprise 8 well-known and 5 novel genes (not documented in COSMIC cancer gene census) based on recurrence, functional impact and cancer cell fraction (CCF). In the revised manuscript, we have reorganized our findings in the table of driver genes in Supplementary Table 3.

Minor comments:

1. What about the variant allele frequencies of the top mutated gene? Any useful comparison with the two other cohorts?

We thank the reviewer for this comment. We had actually calculated cancer cell fraction (CCF), which is VAF but corrected for copy number of the locus and tumour purity (please see Methods section ‘Clonality of somatic variants’ for more details), for all driver mutations in all tumors as part of the driver detection computational analysis. However, we had inadvertently overlooked the inclusion of the requested comparison in the manuscript. We have now included this in Supplementary Fig. 4 and lines 114-117.

2. Full name of WGD should be first shown in page 10.

We thank the reviewer for this correction and have done so.

3. What about characteristics of mutations in oncogenic signaling pathways?

A recent pan-cancer study looked at ten canonical pathways, namely cell cycle, Hippo, Myc, Notch, Nrf2, PI-3-Kinase/Akt, RTK-RAS, TGF β signaling, p53 and β -catenin/Wnt¹. We have now looked at these ten canonical pathways and assessed the distribution of mutations in these pathways (Supplementary Figs. 7-9). Although no pathway was significantly enriched in a particular group, we did observe a significant positive cline of mutation recurrence in the HIPPO pathway from White to Black to Nigerian groups (proportion trend test $P=1.7 \times 10^{-5}$). See Supplementary Fig. 10 and lines 131-137.

4. Though GATA3 and TP53 mutations were found to be almost mutually exclusive, how about the distribution among cohorts or molecular subtypes.

We thank the reviewer for this helpful comment. *GATA3* mutation is virtually absent in the other two cohorts (Fig. 1 and Supplementary Fig. 4) and such analysis was not informative. We have now also added Supplementary Fig. 29 to illustrate the distribution of *GATA3* and *TP53* mutations among cohorts and subtypes.

5. It would be better to use bar plot or pie plot to present information about molecular subtypes between cohorts

Thank you for this helpful suggestion. We have now added bar plots to present the clinical subtypes (ER/PR/HER2) and molecular subtypes (PAM50) across Nigerians and TCGA cohorts (Supplementary Fig. 1).

Reviewer #2, expert in breast cancer genomics and mutational signatures (Remarks to the Author):

Major concerns.

First, there seems to be a bias in all three cohorts. In all three cohort only minority of included cases suffers from HR+/HER2neg disease while this in general is het most common subtype both in Caucasians as well as in African people. For now the bias in the cohorts could be an explanation of the differences observed. Also the TCGA reference cohort is small. If the author would have included 560 ICGC genomes (mostly caucasian, if I am not mistaken) the reference cohort would be more representative and the power of the study much larger. In any case the bias requires an explanation. The Nigerian cohort was 40% HER2 and other African cohort had zero. Are these representative percentage for these populations?

We agree about the limitation of all currently available WGS datasets because none truly represent unbiased population sampling. Clinical subtype (ER/PR/HER2 status) information is much more complete for Nigerian samples in the current study, while only about half of the Nigerian tumors have RNA-seq available for PAM50 subtype classification because the study was funded as an R01 with a budget cap compared to funds available to TCGA and ICGC. Nonetheless, we have implemented a stringent clinical subtype determination and described it in detail in the Methods: “IHC on ER, PR, and HER2 were performed centrally in Nigeria and further reviewed in the United States. Cases with discordant results were again reviewed and resolved by the study pathologists. IHC scoring variables for Allred scoring algorithm were captured according to the 2013 ASCO/CAP standard reporting guidelines. Briefly, for ER and PR testing, immunoreactive tumor cells <1% was recorded as negative and those with $\geq 1\%$ were reported positive. All the positive ER and PR cases were graded in percentages of stained cells and further scored in line with the Allred scoring system.” The field adopted the low cutoff (1%) to enlarge the potential benefits to patients for tamoxifen treatment. With rigorous training and quality control, use of standardized IHC staining and scoring guidance, our Nigerian pathologists have consistently provided ‘clinical grade’ breast tumor subtype information of the utmost quality². There is a high prevalence of HR- tumors in Nigerians, and our current study is consistent with what we and others have previously reported^{3,4}. We specifically did not select cases to be sequenced based on subtype status.

While we agree that larger datasets would have given us more power in identifying differential signals; however, with the White TCGA and Nigerian cohorts analyzed (n=46 and 97 respectively), we were still able to detect *GATA3* and *ZNF217/SYPL1* enrichment in the Nigerians. The TCGA cohort was used for comparative analysis mainly due to data accessibility (availability of raw FASTQs) because we wanted to process all Nigerian and non-Nigerian

samples using a uniform pipeline. There is paucity of African breast cancer genomes and this study will now provide the largest dataset of African breast cancer genomes for future comparative analysis.

We very much appreciate the reviewer's astute observation regarding over-representation of HER2+ in Nigerians. This is indeed a strength of this study. It is known that TCGA data was assembled mainly from eligible convenience tumors from hospitals with unique sample selection criteria. In the revised manuscript, we added population-based data from SEER (Surveillance, Epidemiology, and End Results) and cohort-based data from CBCS (Carolina Breast Cancer Study) in comparison of subtype distribution across cohorts (Supplementary Fig. 1). It is worth noting that because of changing practice patterns, most HER2+ patients in the US are treated with neoadjuvant chemotherapy. As a result, treatment-naïve HER2+ primary tumors in biospecimen banks are often not large enough to be included in genomic studies and do not proportionally reflect population prevalence of this subtype. On the contrary, access to neoadjuvant chemotherapy is limited for Nigerian women with HER2+ disease. The proportion in Nigeria therefore likely represents the 'true' population prevalence of unscreened and untreated HER2+ tumors which is enriched in younger Nigerian women. We also agree that it is difficult to entirely rule out the possibility that we oversampled HER2+ tumors. However, hypothetically, even if we oversampled HER2+ individuals from Nigeria, the comparisons within HER2+ tumors are still valid and incredibly important to the field. Statistical hypothesis testing for racial/ethnic differences was either performed within subtypes or corrected/adjusted for subtype.

Second, if there is not a selection bias, another explanation would be that negative ER status reported was false negative in certain percentage of cases. This could explain why GATA3 mutants are seen among the ER negative cases and while in fact in general them being part of the ER positive cases. The fact that GATA3 is frequently inactivated in triple negative breast cancer (TNBC) is novel and interesting but also unexpected and remarkable. GATA3 is not expressed in the basal layer of the mammary epithelium if I am not mistaken. Inactivation of GATA3 in that cell type will thus not drive tumorigenesis. Thus, if true, the proposed GATA3 inactivation is occurring in a luminal cell type causing it to develop in an ER negative tumor which is possible. This observation, however, demands for independent confirmation and using an orthogonal method. Staining tumors and appropriate controls may help to support the finding but does not fully falsify the finding. I am also interesting in the type of GATA3 mutations observed in the TNBC of this cohort (data not excluded in the submitted manuscript or I overlooked them). In luminal BC, and reported by many groups, the GATA3 mutations are always out of frame mutations and occur in the latter 2 exons) and occur in a specific domain (last zinc finger) of the protein and occur on one allele. This is more in line with a dominant phenotype than the calculated tumor suppressor role here. Tumor suppressor mutations in a gene are wide spread and not localized. The GATA3 mutations in these TNBC cases are expected to be different at least else I find it hard to explain why they occur in TNBC.

In this Life History study, we have made a real discovery by finding that early clonal *GATA3* mutations are associated with a 10.5-year younger age at diagnosis. Thus, we propose that in some patients, if not all, the early clonal *GATA3* mutations might occur in ER- progenitor cells

that happen to be enriched in women of African ancestry in Nigeria, a previously understudied population.

In response to the concern about quality control of our IHC staining, we submit that this bias is based on the assumption of poor quality assurance from our field pathology laboratory in Nigeria. We previously published on this, and we continue to provide pathologists at four academic centers in Nigeria with high quality reagents and continuous quality improvement training as detailed in Oluwasola AO, *et al.*². As explained above, ER/PR/HER2 status have been carefully appraised and cautiously interpreted. In addition, as presented in our previous publication in Nature Communications (Supplementary Figs. 11 and 12 in Pitt JJ, *et al.* 2018)⁴, we compared the ER/PR/HER2 status with *ESR1/PGR/ERBB2* gene expression in Nigerian samples with mRNA expression data available. Overall, IHC calls corroborated by *ESR1*, *PGR*, and *ERBB2* expression. We also previously published IHC results with excellent quality control showing overrepresentation of aggressive subtypes in women from Nigeria and Senegal³. We hope that references to these prior publications alleviate the reviewer's concern on ER false negativity. Furthermore, we went back to restain 16 leftover Nigerian breast tissues with ER-status and purity $\geq 60\%$. We observed positive vimentin staining, which indicates that our tissue quality is good for antibody staining. In addition, GATA3 staining intensity varies from 1 to 3 (average 2.38) in these 16 ER- tumors, confirming breast epithelial origin of the samples. Among them, one *GATA3* mutated tumor had GATA3 staining intensity of 3, which supports the previous finding that *GATA3* gene/protein expression is generally higher in *GATA3* mutated tumors compared with wild-type tumors.

It is true that *GATA3* somatic mutations are often found in ER+ breast tumors and they are rarely seen in ER- tumors from studies recruiting women of European ancestry predominantly. The literature is filled with data from European ancestry groups but the paucity of data from non-European ancestry groups hampers making definitive conclusions about the significance of *GATA3* mutations in women of African ancestry, whose genomes are organized differently from women of European ancestry. About 10% of African genome was described as missing from the reference genome⁵ underscoring the importance and significance of our findings to breast cancer research. In Nigerian tumors, 20 *GATA3* mutations were observed – 10 in ER+ and 10 in ER- tumors (3 HR+/HER2-, 10 HER2+ and 7 HR-/HER2-). Although it is known that the concordance between IHC and PAM50 subtypes is about 60-70%, for Nigerian tumors with both WGS and RNA-seq data available, we looked at the PAM50 subtypes. Half (n=10) of the 20 tumors harboring *GATA3* mutations had PAM50 molecular subtypes -- 5 ER+ tumors are assigned as LumA (n=3) and LumB (n=2), while 5 ER- tumors were determined to be HER2 (n=3), LumB (n=1) and Basal (n=1). We were not able to solidly conclude a genotype-phenotype correlation of *GATA3* mutation in ER- tumors, due to the small sample size. However, this is a real discovery that opens up the field for further investigations.

The literature on the role of *GATA3* mutations in ER- tumors is a question worthy of further exploration with a series of comprehensive experiments that could result in separate publications. Nonetheless, we gathered evidence below to support our conjecture that the existence of *GATA3* mutations in ER- tumors is plausible.

In breast tumors overall, *GATA3* mutations correlated positively with *GATA3* gene expression, which is correlated positively with *GATA3* protein expression. We looked into *GATA3* expression in different cancers in Pan-Cancer Atlas at cbiportal.org. *GATA3* expression is higher in bladder cancer, miscellaneous neuroepithelial tumor, and pheochromocytoma that are not estrogen-related. On the other hand, in other female cancers like endometrial carcinoma and ovarian epithelial tumors, *ESR1* expression level is relatively high but *GATA3* expression level is not. In addition, except breast cancer (and endometrial cancer), *GATA3* mutations do not correlate to *ESR1* expression (e.g. in non-small cell lung cancer, esophagogastric adenocarcinoma and colorectal adenocarcinoma). These observations indicate that *GATA3* may play a different role in pathway(s) beyond ER-signaling.

GATA3 has been reported to play a major role in cell lineage specification and development in a variety of cells, tissues, and organs and it has been implicated in several cancer types. “It often acts at the level of stem and progenitor cells in which it controls the expression of key lineage-determining factors as well as cell cycle genes, thus *GATA3* as a transcription factor is one of the main drivers of cell fate choice and tissue morphogenesis.”⁶ Immunofluorescence microscopy analysis revealed substantial *GATA3* expression in ER- cells⁷. Previous work demonstrated that the luminal epithelial cells express *GATA3* while the myoepithelial cells express very low levels of *GATA3*⁸. Recent studies showed that not all the luminal cells are ER+ -- ER+ and ER- progenitors give rise to ER+ and ER- differentiated cells, respectively. “Ductal and alveolar hormone-receptor negative progenitors are distinct lineages and there is also a separate hormone receptor positive luminal lineage”⁹.

Moreover, one cannot rule out the possibility that those *GATA3* mutations occurred in ER+ luminal cells but later those cells became ER-, because of the dynamic nature of ER status. For example, it has been shown that hyperactive GFR (growth factor receptor) signaling can be a molecular determinant of ‘ER loss’ because ER can crosstalk directly or indirectly with and up-regulate various GFR tyrosine kinases such as HER2¹⁰. Whether this can in part explain our finding that *GATA3* mutations occur in ER-/HER2+ tumors remains unknown.

Based on our computational analyses (cDriver and 20/20 principle), *GATA3* was suggested to be a TSG, but we acknowledge that contradictory functions of *GATA3* as either TSG and/or ONC in breast cancer has been described in the literature¹¹. We completely agree with the notion that “as the catalog of oncogenic driver mutations is expanding, it becomes clear that alterations in a given gene might have different functions and should not be lumped into one class. The transcription factor *GATA3* is a paradigm of this.”¹² Functional characterization of each *GATA3* mutation is beyond the scope of the current work, but we agree that further studies on *GATA3* individual mutations in the context of tumor progression (see response to Reviewer 1’s second comment) are warranted. In addition, we thank the reviewer for the suggestion of examining the mutation localization. Given the small number of *GATA3* mutations found in Nigerian tumors (dividing mutations among subtypes yields very small numbers), we provided a series of *GATA3* mutation lollipop plots across subtypes and cohorts in Supplementary Figs. 5 and 6.

Third. Mutual exclusivity between TP53 and *GATA3* is reported. This can easily be explained by them being present in a different subtype. This is what is seen in causation and all reported

cohorts thus far. But if indeed present in TNBC one might argue that they occur in different TNBC subtypes. Since there is RNAseq for a subset could this shed light on this.

Thank you for this thoughtful suggestion. In accordance with Reviewer 1's suggestion, we added plots to illustrate the distribution of *GATA3* and *TP53* mutations among cohorts and subtypes (Supplementary Fig. 29). We investigated the PAM50 molecular subtypes along with clinical subtypes (ER/PR/HER) for 20 Nigerian *GATA3* mutation carriers. We did not gain a clear view of genotype-phenotype correlation, very likely due to the small sample size. However, from a Life History and evolutionary point of view, given the early clonal nature of both drivers, this mutual exclusivity is likely to be a genomic representation of alternative tumor trajectories in Nigerian breast cancers. The enrichment of the *GATA3*-initiating trajectory in Nigerians and its association with younger age at diagnosis may explain to some extent the ethnic disparities in breast cancer survival. This deserves further exploration in larger datasets from Africa where the risk factors for young onset breast cancer are yet to be discovered.

Fourth, signatures are called using non-negative factorization (nnF) which makes sense and using the newest algorithm of Alexandrov. However, using limited sized cohorts there is a risk of mis-classification. Did the author also pursue *de novo* calling of signatures. This does not force mutations in upfront defined classes and might provide independent confirmation if SBS39 is correctly called or not. Some is true for other novel signatures. Including larger cohorts could help here as well.

We did indeed use *de novo* calling of signatures, and we did include additional samples for mutational signature calls to increase our sample set. Specifically, for the SNV/INDEL mutational signature analysis, we included variant calls on additional breast cancer samples from the Pan-Cancer Analysis of Genomes (PCAWG; mainly White European ancestry) dataset to have more power in *de novo* extraction of signatures. Please refer to our description in Methods: "De-novo extraction and decomposition to known cosmic mutation signatures in single base substitution (SBS), double base substitution (DBS), as well as small insertion and deletion (ID) formats were implemented based on a non-negative matrix factorization (NNMF) framework using SigProfilerExtractor" "Because NNMF is more accurate with a larger number of samples, we increased our sample set by adding 128 additional breast cancer WGS samples from the Pan-cancer Analysis of Whole Genomes (PCAWG) study and eight TCGA (Asian and unassigned ancestries) WGS samples. Our final input dataset for SigProfilerExtractor thus included a total of 309 samples."

Fifth, many differences (drivers, CNAs, heterogeneity, etc) between groups (black (Nigerian or other) and white) are not corrected for subtype. One cohorts (non-Nigerian Africans) lack even a subtype. This precludes proper comparison particularly in the subgroup being absent. It does not become clear how this was corrected for (apart from the bias).

We thank the reviewer and agree that adjusting for subtype is important for proper comparison. We had taken into account subtype and corrected for this in some comparisons (e.g. line 108 and line 264). In the case of CNA (i.e. 14q LOH), we have only compared groups within a specific

subtype (HER2+) where enrichment has been observed. For mutational drivers, only *GATA3* was found to be enriched in Nigerians. This association (original $P=0.0013$) is now adjusted for subtype and remains significant post-correction (generalized linear model $P=0.0032$; see lines 123-125). For heterogeneity, we had indeed included clinical subtype in the generalised linear model along with number of reads per chromosome copy (NRPCC; an accurate measure of effective sequencing depth) but apologise for inadvertently failing to mention this in the manuscript. This has now been corrected (please see line 287).

HRD is defined based on signature 3 only. This is a poor classifier of HRD. Using additional information such as indel size and number, certain rearrangements (tandem duplications) etc as defined by Nik-Zainal and coworkers (HRDirect) or Cuppen E and coworkers (Chord) are superior. Why were these not used to classify HRD.

We appreciate this valuable suggestion and agree with the reviewer. In the updated analyses, structural variants have been called and HRD scores were calculated using CHORD (Supplementary Fig. 18) -- please see updated Results (lines 199-204) and updated Methods (lines 572-602). In addition, for HRD, we have an additional layer of information from the recently-developed INDEL mutational signature analysis (SigProfilerExtractor)¹³ which showed a strong correlation between NHEJ signatures (ID6+ID8) and CHORD HRD score ($R=0.93$, $P<2.2\times 10^{-16}$) indicating further evidence for presence of HRD (Supplementary Fig. 19).

I wonder whether Nigerian associated CNAs (e.g. 14q LOH) hold if larger control cohorts are included which are widely available.

We acknowledge that larger cohorts would provide more power in differential analysis. Since 14q LOH was specifically enriched in the HER2+ subtype, we have now compared the Nigerian cohort with the entire PCAWG breast cancer dataset. For this, we identified HER2+ PCAWG samples (total $n=25$) and compared the prevalence of 14q LOH in Nigerian patients with patients of White European descent ($n=22$; ~90%). We observed a significant enrichment in the Nigerian cohort ($P=0.0034$), corroborating our previous finding. We now mention this in the text in lines 234-237.

Also more kataegis spots are observed in Nigerian cases. However, the Nigerian cohort is HER2 enriched and APOBEC mutagenesis high in HER2 cases. So will this hold in better matched groups or when corrected for subtype.

We thank the reviewer for this comment. We did undertake subtype adjustment in our kataegis analysis and noticed that the rate of kataegis foci only remains significantly higher when compared with the White group. Based on an additional comment below, we now also adjust for differences in sequencing depth as measured by the number of reads per chromosome copy (NRPCC). Please see lines 264-265.

I am not sure why the comparisons was not also compared to the ICGC cohort involving 560 breast cancers (Zainal et al). Particularly the significant findings. I am not sure whether ancestry is reported for all cases in that cohort but it will increase power to find significant differences.

Please see our response above.

I would not use PGA since it is generally known to be inferior to the number of aberration in a genome when calling genomic instability as it ignores events being often independent.

Thank you for this comment. We have now removed PGA results and focus on PGAn which incorporates the number of aberrations. Please see lines 253-257.

Minor points.

I would not use black and white but African versus Caucasian. It is also not defined where the non-Nigerian Africans originate from. This should be made more clear. Are the African Americans or non-Nigerian Africans.

Thank you but African versus Caucasian is incorrect and not applicable to this study. We defined these three groups in the first paragraph of Results: "...Nigerian Black (Nigerian for short, n=97), White TCGA (White for short, n=46) and Black TCGA (Black for short, n=30) groups...". We used the same terms in our previous publications in JAMA Oncology¹⁴ and Nature Communications⁴ for consistency.

HER2 cases in the Nigarian cases was high. Was this also technical are was this confirmed in the WGS data.

We did confirm HER2 status with copy number calls by sequencing. Please refer to Methods: "IHC on ER, PR, and HER2 were performed centrally in Nigeria and further reviewed in the United States. Cases with discordant results were again reviewed and resolved by the study pathologists. IHC scoring variables for Allred scoring algorithm were captured according to the 2013 ASCO/CAP standard reporting guidelines.... Percentage of tumor staining for HER2 test were also reported along with a score of 0 and 1+ as negative, 2+ as equivocal, and 3+ as positive case. HER2 equivocal cases were further confirmed with genomic copy number calls."

Regarding the higher proportion of HER2+ tumors in the Nigerian samples, please see our response above.

Was the sequence depth between cohort the same and the sequencing of the same quality. This might affect for instance the number of Kateagis foci called as well as the clonality scores.

We thank this reviewer for this observation. We processed the Nigerian and TCGA tumors and matched normals uniformly and conducted rigorous variant calling and quality check (see

Methods). Whole-genome sequencing depth was indeed different between Nigerian and TCGA WGS tumor/normal (T/N) pairs where the former were sequenced at a higher depth -- Nigerian: 103.2x/35.1x versus TCGA: 52.3x/33.5x. Similar to the intra-tumoral heterogeneity analysis where we adjust for this using the number of reads per chromosome copy (NRPCC), we re-examined the difference in the number of kataegis foci after adjusting for clinical subtype and NPRCC in a generalised linear model. After adjustments, the higher rate of kataegis foci only remained statistically significant when compared with the White group (please see lines 264-265).

RNA data are not provided so they can be deleted from the manuscript. Unless may be used to independently confirm phenotype.

We performed differential gene expression analysis between Nigerian quiet genomes vs. *GATA3/TP53* positive genomes -- please refer to the last paragraph of the Discussion and Supplementary Figs. 33 and 34 as well. Among 97 Nigerian WGS tumors, 47 had RNA-seq data available, we employed mRNA expression-based PAM50 subtype classification in the revised manuscript.

Reviewer #3, expert in breast cancer populations (Remarks to the Author):

This is a very interesting analysis of somatic tumor genetics in a understudied population. Specific comments are below:

There was a lot of work investigating HRD but some of this could be improved noting the below points

1) Did the authors run a rearrangement signature analysis similar to determine if *BRCA1/2* are correlated with the signatures in this paper <https://www.nature.com/articles/nature17676>

We thank the reviewer for this query. We have now called structural variants and ran SV signature analysis on the entire cohort -- please see updated Results (lines 199-204) and updated Methods (lines 572-602). We detected all six signatures in the cited article¹⁵ and were able to examine correlation of these signatures with *BRCA*-type. We observed a strong correlation between a) *BRCA1* and SV signature S3 and b) *BRCA2* and SV signature S2 (Fig. 4 and Supplementary Fig. 19).

2) Similarly, with the *BRRCA1/2* HRD signature are they *BRCA1*-like or *BRCA2*-like ? <https://www.nature.com/articles/s41467-020-19406-4> or HRDetect (<https://www.nature.com/articles/nm.4292>) from A. Degasperi et al. A practical framework and online tool for mutational signature analyses show intertissue variation and driver dependencies, Nature Cancer, [<https://doi.org/10.1038/s43018-020-0027-5>], 2020.

We thank this reviewer (and also Reviewer 2) for suggesting calculating HRD scores as part of the analysis. By using CHORD, we identified HRD tumors and predicted *BRCA1/2*-type tumors.

We observed a complete match between *BRCA* mutations and predicted *BRCA*-type in all *BRCA*-positive tumors. This classification also matched completely with the SV signature-based inference of *BRCA*-types. We are now dedicating a separate subsection to HRD which now includes the additional CHORD and SV signature results. This has vastly improved the manuscript as presented in Fig. 4, which now informs the major conclusion from our study.

3) DO the authors know the biallelic status of *BRCA1/2* and if biallelic have a HRD signature but not monoallelic?

We inspected the relationship between allelic status of *BRCA1/2* and HRD signature. Supplementary Table 4 contains all information on *BRCA*-positive samples including the type and origin of mutations along with copy number status at *BRCA* loci. Of the 15 *BRCA*-positive samples, eight were unequivocally biallelic, one unequivocally monoallelic and six equally likely to be mono- or bi-allelic due to having a germline mutation along with a somatic copy loss. Assuming ambiguous samples as mono-allelic, there was no significant difference in HRD score between bi-allelic and mono-allelic status (0.89 vs. 0.86, $P=0.26$) with scores ranging from 0.85-0.94 and 0.78-0.94 respectively.

4) Can the authors clarify if the hotspot non-coding analysis is somatic or germline? I suspect somatic but maybe useful to explain in the legend of Figure 2 as they note variants. In general, the legend will benefit from more details to make it clearer to readers

They are somatic non-coding variants, and we have updated Fig. 2 legend as suggested. Thank you.

5) *ZNF217* non-coding hotspot found in 40% of AA compare to 4% in whites? are there confidence intervals for this estimate and does gene expression data in TCGA support non-coding mutations are related to altered levels of expression in AA?

Thank you for the suggestion. We have now added the 95% CI for the frequencies reported for the Nigerian and White group estimates in the manuscript (line 142). We aimed to analyze the gene expression of *ZNF217* and *SYPL1* between Nigerians and White TCGA; however, the number of tumors with these mutations were 2 and 0 in White TCGA respectively and no informative comparison could be made. We, nonetheless, did compare the mutant and wild-type Nigerian and TCGA Black samples for these two genes which had RNA-seq data available. We did observe a generally elevated expression level in both the *ZNF217* and *SYPL1* genes for mutant samples but not statistically significant (Supplementary Fig. 11). Since the functional consequences of the identified non-coding mutations are yet to be established, they may result in either up- or down-regulation of these genes and may not show a clear up- or down-regulation expression pattern as a group. We now refer to this additional analysis in the manuscript (see lines 152-157).

6) The samples used from the study are a subset of the larger Nigerian study population how representative are these samples from the overall study in terms of risk factors/tumor characteristics?

We much appreciate all three reviewers' suggestions to better present the patient/tumor characteristics. We updated the supplementary table and also added bar charts for the distribution of tumor subtypes across study cohorts. Please also refer to our response to Reviewer 2's first major concern above.

In addition, we agree that both TCGA and this study used convenient samples ascertained in Hospitals and may not reflect population rates. We add this statement as one of the limitations in the Discussion.

References

- 1 Sanchez-Vega, F. *et al.* Oncogenic Signaling Pathways in The Cancer Genome Atlas. *Cell* **173**, 321-337 e310, doi:10.1016/j.cell.2018.03.035 (2018).
- 2 Oluwasola, A. O. *et al.* Use of Web-based training for quality improvement between a field immunohistochemistry laboratory in Nigeria and its United States-based partner institution. *Ann Diagn Pathol* **17**, 526-530, doi:10.1016/j.anndiagpath.2013.07.003 (2013).
- 3 Huo, D. *et al.* Population differences in breast cancer: survey in indigenous African women reveals over-representation of triple-negative breast cancer. *J Clin Oncol* **27**, 4515-4521, doi:10.1200/JCO.2008.19.6873 (2009).
- 4 Pitt, J. J. *et al.* Characterization of Nigerian breast cancer reveals prevalent homologous recombination deficiency and aggressive molecular features. *Nat Commun* **9**, 4181, doi:10.1038/s41467-018-06616-0 (2018).
- 5 Sherman, R. M. *et al.* Assembly of a pan-genome from deep sequencing of 910 humans of African descent. *Nat Genet* **51**, 30-35, doi:10.1038/s41588-018-0273-y (2019).
- 6 Zaidan, N. & Ottersbach, K. The multi-faceted role of Gata3 in developmental haematopoiesis. *Open Biol* **8**, doi:10.1098/rsob.180152 (2018).
- 7 Asselin-Labat, M. L. *et al.* Gata-3 is an essential regulator of mammary-gland morphogenesis and luminal-cell differentiation. *Nat Cell Biol* **9**, 201-209, doi:10.1038/ncb1530 (2007).
- 8 Chou, J., Provot, S. & Werb, Z. GATA3 in development and cancer differentiation: cells GATA have it! *J Cell Physiol* **222**, 42-49, doi:10.1002/jcp.21943 (2010).
- 9 Cristea, S. & Polyak, K. Dissecting the mammary gland one cell at a time. *Nat Commun* **9**, 2473, doi:10.1038/s41467-018-04905-2 (2018).
- 10 Lopez-Tarruella, S. & Schiff, R. The dynamics of estrogen receptor status in breast cancer: re-shaping the paradigm. *Clin Cancer Res* **13**, 6921-6925, doi:10.1158/1078-0432.CCR-07-1399 (2007).
- 11 Takaku, M., Grimm, S. A. & Wade, P. A. GATA3 in Breast Cancer: Tumor Suppressor or Oncogene? *Gene Expr* **16**, 163-168, doi:10.3727/105221615X14399878166113 (2015).
- 12 Hruschka, N. *et al.* The GATA3 X308_Splice breast cancer mutation is a hormone context-dependent oncogenic driver. *Oncogene* **39**, 5455-5467, doi:10.1038/s41388-020-1376-3 (2020).
- 13 Alexandrov, L. B. *et al.* The repertoire of mutational signatures in human cancer. *Nature* **578**, 94-101, doi:10.1038/s41586-020-1943-3 (2020).
- 14 Huo, D. *et al.* Comparison of Breast Cancer Molecular Features and Survival by African and European Ancestry in The Cancer Genome Atlas. *JAMA Oncol* **3**, 1654-1662, doi:10.1001/jamaoncol.2017.0595 (2017).
- 15 Nik-Zainal, S. *et al.* Landscape of somatic mutations in 560 breast cancer whole-genome sequences. *Nature* **534**, 47-54, doi:10.1038/nature17676 (2016).

Reviewers' Comments:

Reviewer #1:

Remarks to the Author:

The authors have addressed the reviewers' concerns.

Reviewer #3:

Remarks to the Author:

No further comments

Response to reviewers
Ansari-Pour N & Zheng Y *et al.* - NCOMMS-20-49388A

REVIEWER COMMENTS

Reviewer #1 (Remarks to the Author):

The authors have addressed the reviewers' concerns.

We thank the reviewer for the careful review.

Reviewer #3 (Remarks to the Author):

No further comments

We thank the reviewer for the careful review.